# MoM: Linear Sequence Modeling with Mixture-of-Memories

**Jusen Du[1,2]**[*], **Weigao Sun[2]**[†§], **Disen Lan[2,3]**[*], **Jiaxi Hu[4]**, **Tao Zhang[1]**[†], **Yu Cheng[5]**[†]

[1]Tsinghua University, [2]Shanghai AI Laboratory, [3]Fudan University,
[4] The Hong Kong University of Science and Technology (Guangzhou),
[5]The Chinese University of Hong Kong

## Abstract

Linear sequence modeling methods, such as linear attention, state space modeling, and linear RNNs, offer significant efficiency improvements by reducing the complexity of training and inference. However, these methods typically compress the entire input sequence into a single fixed-size memory state, which leads to suboptimal performance on recall-intensive tasks. To address this limitation, we introduce a novel architecture called Mixture-of-Memories (MoM). MoM utilizes multiple independent memory states, with a router network directing input tokens to specific memory states. This approach greatly enhances the overall memory capacity while minimizing memory interference. MoM serves as a general framework that can be seamlessly combined with diverse memory update mechanisms across linear models. As a result, MoM performs exceptionally well on recall-intensive tasks, surpassing existing linear sequence modeling techniques. Despite incorporating multiple memory states, the computation of each memory state remains linear in complexity, allowing MoM to retain the linear-complexity advantage during training, while constant-complexity during inference. Our experimental results show that MoM outperforms current linear sequence models on downstream language tasks, particularly recall-intensive tasks, and even achieves performance comparable to Transformer models.

## 1 Introduction

Attention mechanisms have made significant contributions to the field of artificial intelligence, advancing various modalities such as language, vision, audio, video, graphs, and even time series (Achiam et al., 2023; Team, 2023). The Transformer (Vaswani, 2017), known for its ability to capture long-range dependencies, has become a foundational architecture in this space. However, traditional Transformers encounter computational challenges due to their quadratic time complexity, $O(n^2)$, with respect to sequence length $n$, making it difficult to scale to long sequences. To overcome this limitation, several linear sequence modeling methods have been proposed, including linear attention (Katharopoulos et al., 2020; Qin et al., 2023a; Li et al., 2025), state space modeling (Gu & Dao, 2024; Dao & Gu, 2024), and linear RNNs (Peng et al., 2024; Qin et al., 2024d), which offer $O(n)$ training complexity and $O(1)$ inference complexity. These approaches often reduce the input sequence to a fixed-size hidden space, collapsing the information into a single "memory state". While these methods enhance efficiency, they face two main challenges: **limited memory capacity** and **memory interference**. When new information overwrites the single fixed-size memory state, previously stored representations may degrade, which negatively impacts its long-term memory performance on recall-intensive tasks.

We argue that the strong performance of Transformer models on recall-intensive tasks arises from their ability to avoid memory interference by maintaining independent key-value caches for each token, thus offering virtually unlimited memory capacity. In contrast, linear sequence modeling relies on extreme compression, consolidating all the input information into a single fixed-size memory state (Katharopoulos et al., 2020; Dao & Gu, 2024). This approach results in limited memory capacity and inherently leads to memory interference issues.

---

[*]Interns at Shanghai AI Laboratory; [†]Corresponding Authors; [§]Project Lead.

Interestingly, the human brain has developed mechanisms that enable large memory capacity while reducing memory interference. Neuroscience studies show that in the hippocampus, theta oscillations (4~8 Hz) and gamma oscillations (30~100 Hz) work together to support a neural coding mechanism for multi-item memory (Buzsáki, 2002; Lisman & Jensen, 2013). Specifically, each theta cycle is subdivided into multiple gamma subcycles, and within each gamma subcycle, a distinct group of neurons is activated following the "E%-max" mechanism (de Almeida et al., 2009). This sequential activation temporally separates different memory items, thus preventing interference.

Inspired by these biological insights, we propose a new architecture called **Mixture-of-Memories (MoM)**, which aims to strike a balance between the explicit token representations in Transformers and the extreme compression found in earlier linear sequence modeling methods. MoM employs multiple independent memory states, with a router network that directs input tokens to specific memory states. The input sequence is divided into a predefined number of subsequences (phase-specific neural assemblies), which are processed in parallel and fed into the corresponding memory projections (dentate microcircuits) to generate key-value pairs. As the linear sequence modeling layer processes each subsequence using an RNN-like update mechanism, it produces multiple memory states that capture different aspects of the input sequence. The final output is computed as a weighted sum of these memories, which we refer to as the mixture-of-memories. This approach expands memory capacity and eliminates memory interference, enabling MoM to significantly outperform existing linear sequence models that rely on a single fixed-size memory state.

Our contributions can be summarized as follows:

- We present MoM, an architecture that incorporates multiple independent memory states, significantly enhancing memory capacity and eliminating memory interference, while retaining the efficiency benefits of linear-time training and constant-memory inference.
- Distinct with existing gating mechanisms, MoM is a new paradigm to reduce memory interference by separating the memory states. The overall design is broadly compatible with diverse linear sequence modeling methods, making it a straightforward and effective approach to boost task performance.
- Through empirical evaluation, we show that MoM outperforms strong linear sequence modeling baselines across a variety of language tasks, particularly on recall-intensive tasks. MoM even achieves performance on par with Transformer models, a feat that current linear sequence modeling methods struggle to match.

## 2 PRELIMINARY

For notations in this work, we use bold lower-case letters for row vectors (e.g., $\boldsymbol{q}_t, \boldsymbol{k}_t$), bold upper-case letters for matrices (e.g., $\boldsymbol{Q}, \boldsymbol{K}$) and the identical letters represent a row in the matrix, e.g., $\boldsymbol{q}_t$ is the $t$-th row of $\boldsymbol{Q}$.

### LINEAR ATTENTION

To reduce the time complexity of Transformer attention, various optimization techniques have been proposed. Linear Transformers (Katharopoulos et al., 2020) replace the softmax attention mechanism with dot-product of feature maps $\phi(\cdot)$:

$$\boldsymbol{o}_t = \frac{\sum_{i=1}^{n} \phi(\boldsymbol{q}_t)\phi(\boldsymbol{k}_i)^T \boldsymbol{v}_i}{\sum_{i=1}^{n} \phi(\boldsymbol{q}_t)\phi(\boldsymbol{k}_i)^T}, \tag{1}$$

where $\boldsymbol{q}_t, \boldsymbol{k}_t, \boldsymbol{v}_t \in \mathbb{R}^d$. While the presence of the denominator may lead to numerical instability (Qin et al., 2024b) and the feature map can utilize an identity function, which we omit for simplicity. In perspective of memory, the formulation can also be written in a recurrent format:

$$\boldsymbol{M}_t = \boldsymbol{M}_{t-1} + \boldsymbol{k}_t^T \boldsymbol{v}_t, \quad \boldsymbol{o}_t = \boldsymbol{q}_t \boldsymbol{M}_t. \tag{2}$$

This indicates that linear attention can function as a linear recurrent layer with a matrix-valued hidden state $\boldsymbol{M}$ which we refer to as memory sate and the output is generated by querying the memory state $\boldsymbol{M}$. This represents the ultimate compression of sequence information, condensing the entire sequence into a single memory state.

Building on the foundational concepts of linear attention and memory perspective, some recent advancements have focused on optimizing memory structure, including gated updates (Yang et al., 2023; Qin et al., 2024e;d) and memory capacity expansion (Peng et al., 2024; Qin et al., 2024d).

# 3 METHOD

## 3.1 MOTIVATION

Linear sequence models compress the entire sequence data into a fixed-size memory state. Despite numerous efforts to minimize information loss, such as introducing gating mechanisms and employing more precise control over memory modifications (Orvieto et al., 2023; De et al., 2024; Beck et al., 2024; Yang et al., 2023; Zhang et al., 2024), some degradation in this compression process is inevitable. Expanding the memory capacity has been shown to mitigate this issue to some extent, with studies indicating that increasing memory capacity can enhance model performance (Qin et al., 2024d; Peng et al., 2024).

However, previous approaches that simply increased the size of the RNN state, essentially expanding a single memory state, struggled to capture the full spectrum of information within an entire sequence. We propose that this difficulty arises because sequence information is often multifaceted, and a single, expanded memory may not be capable of simultaneously capturing multiple aspects of the data. Inputs that introduce new or orthogonal information may interfere with existing memory content when using a shared memory. Rather than discarding these inputs through gating mechanisms or overwriting the existing memory state, it may be more effective to consider alternative strategies that allow for the preservation of diverse information without interference.

## 3.2 MOM: MIXTURE-OF-MEMORIES

To address the challenge outlined above, we propose a novel approach for encoding multi-item memory such as theta-gamma oscillations (Lisman & Jensen, 2013), and concepts from Mixture-of-Experts (MoE) (Shazeer et al., 2017), where different experts handle specific tokens. In this approach, we leverage multiple memory states, each of which is selectively updated by different inputs. This increases the memory capacity and enables the model to retain diverse pieces of information by storing various types of inputs in separate memory states.

In our framework, the memory states function similarly to the experts in MoE. However, instead of relying on completely separate networks, these modules are individual RNN states embedded within a linear recurrent mechanism. This design allows for the isolation of memory updates while concurrently managing distinct types of information. It is important to note that MoM essentially differs from traditional MoE, as we will discuss in Appendix B. Figure 1 provides an overview of the MoM architecture. Below, we introduce the structure of the MoM layer and explain how this multimemory architecture is implemented in the context of linear sequence modeling.

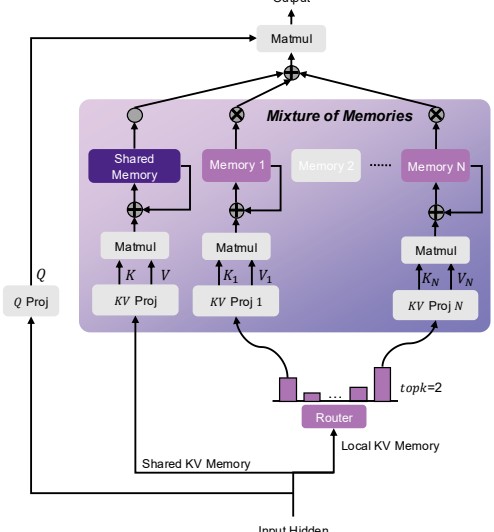

Figure 1: **MoM Architecture.** Each input token selectively activates and updates $K$ memory states, leaving non-activated memory states unchanged to avoid interference from current input. Additionally, we introduce a continuously activated shared memory. This figure presents the basic memory update mechanism; other mechanisms involving gating or more complex updates follow a similar approach.

### 3.2.1 ROUTER

We use a router to assign inputs to different memory states. Utilizing the top-$k$ concept, each token is routed to the top-$k$ memories based on its importance scores. Specifically, we use a simple linear layer to generate these scores for each input token. After applying a softmax function, we select the top-$k$ scores and normalize them.

$$\mathbf{scores}_t = \text{TopK}(\text{softmax}(\boldsymbol{x}_t \boldsymbol{W}_g)) \in \mathbb{R}^k, \tag{3}$$

$$\boldsymbol{g}_t = \frac{\mathbf{scores}_t}{\sum \mathbf{scores}_t} \in \mathbb{R}^k, \tag{4}$$

where $\boldsymbol{x}_t \in \mathbb{R}^d$, $k$ is the top-$k$ number, $\boldsymbol{W}_g \in \mathbb{R}^{d \times M}$ is learnable weight, $\boldsymbol{g}_t$ is the normalized importance scores of the input $\boldsymbol{x}_t$.

### 3.2.2 LINEAR RECURRENT MEMORY MODULE

After the router network, the input $\boldsymbol{x}_t$ is directed to top-$k$ linear recurrent modules, meaning that the top-$k$ memories are activated while the others remain inactive.

**Each Memory.** For each activated memory, indexed by $m$, we perform the following operation:

1. **Key and Value Projections**: We project the input $\boldsymbol{x}_t$ to $\boldsymbol{k}_t^m$ and $\boldsymbol{v}_t^m$ using $\boldsymbol{W}_k^m$ and $\boldsymbol{W}_v^m$:

$$\boldsymbol{k}_t^m = \boldsymbol{x}_t \boldsymbol{W}_k^m, \boldsymbol{v}_t^m = \boldsymbol{x}_t \boldsymbol{W}_v^m \in \mathbb{R}^d, \tag{5}$$

   where $\boldsymbol{W}_k^m, \boldsymbol{W}_v^m$ are learnable projection weights for $kv$ of the $m$-th memory module.

2. **Memory Update**: We update the activated memory state using $\boldsymbol{k}_t^m, \boldsymbol{v}_t^m$:

$$\boldsymbol{M}_t^m = \boldsymbol{M}_{t-1}^m + (\boldsymbol{k}_t^m)^T \boldsymbol{v}_t^m \in \mathbb{R}^{d \times d}. \tag{6}$$

   The equation above represents the simplest form of memory update for clarity. Our approach is flexible and does not rely on a specific memory update mechanism. To enhance performance, we can incorporate mechanisms such as forget gates (Sun et al., 2023).

   More generally, our method can be adapted to incorporate various memory update methods proposed in previous work. Detailed descriptions of these methods are provided in Table 1.

**Memory Mixing.** After updating the activated memory states, we perform a weighted sum of these memory states using the importance scores obtained from Equation(4).

$$\tilde{\boldsymbol{M}}_t = \sum_m g_t^{(m)} \boldsymbol{M}_t^m \in \mathbb{R}^{d \times d}, \tag{7}$$

where $\boldsymbol{M}_t^m$ is one activated memory and $g_t^{(m)}$ is the importance score of $\boldsymbol{M}_t^m$.

We then obtain the output of the MoM by applying query vector $\boldsymbol{q}_t$ to the mixed memory $\tilde{\boldsymbol{M}}_t$:

$$\boldsymbol{o}_t = \boldsymbol{q}_t \tilde{\boldsymbol{M}}_t \in \mathbb{R}^d. \tag{8}$$

Finally, the output of the MoM layer is computed by applying an activation function, normalization, and a linear transformation.

Throughout the recurrent process, only a subset of memory states is activated and updated at each time step, while memory states that are not routed remain inactive and unchanged. When the input passes through the key-value projection layer, it generates multiple sets of keys and values that are fed into different memory modules. This design enables the model to maintain multiple memory states, each preserving distinct pieces of information. By aggregating the activated memories into a comprehensive mixed

Table 1: **Memory Update Rules.** We demonstrate that several linear sequence models can be viewed as recurrent models in terms of memory updates, where $a_t, b_t \in (0, 1)$ are data-dependent scaler, $\boldsymbol{a}_t$ is data-dependent vector, and $\gamma$ is a data-independent constant.

| Method | Memory Update Rule |
|---|---|
| Linear Attn | $\boldsymbol{M}_t = \boldsymbol{M}_{t-1} + \boldsymbol{k}_t^T \boldsymbol{v}_t$ |
| RetNet | $\boldsymbol{M}_t = \gamma \boldsymbol{M}_{t-1} + \boldsymbol{k}_t^T \boldsymbol{v}_t$ |
| GLA | $\boldsymbol{M}_t = (\boldsymbol{a}_t^T \mathbf{1}) \boldsymbol{M}_{t-1} + \boldsymbol{k}_t^T \boldsymbol{v}_t$ |
| DeltaNet | $\boldsymbol{M}_t = (\boldsymbol{I} - \boldsymbol{k}_t^T \boldsymbol{k}_t) \boldsymbol{M}_{t-1} + b_t \boldsymbol{k}_t^T \boldsymbol{v}_t$ |
| G-DeltaNet | $\boldsymbol{M}_t = a_t (\boldsymbol{I} - \boldsymbol{k}_t^T \boldsymbol{k}_t) \boldsymbol{M}_{t-1} + b_t \boldsymbol{k}_t^T \boldsymbol{v}_t$ |
| TTT | $\boldsymbol{M}_t = \boldsymbol{M}_{t-1} + b_t \nabla l(\boldsymbol{M}_{t-1}; \boldsymbol{k}_t, \boldsymbol{v}_t)$ |
| Titans | $\boldsymbol{M}_t = a_t \boldsymbol{M}_{t-1} + b_t \nabla_M l(\boldsymbol{M}_{t-1}; \boldsymbol{k}_t, \boldsymbol{v}_t)$ |
| Mamba2 | $\boldsymbol{M}_t = a_t \boldsymbol{M}_{t-1} + b_t \boldsymbol{k}_t^T \boldsymbol{v}_t$ |
| HGRN2 | $\boldsymbol{M}_t = (\boldsymbol{a}_t^T \mathbf{1}) \boldsymbol{M}_{t-1} + (1 - \boldsymbol{a}_t)^T \boldsymbol{v}_t$ |
| RWKV6 | $\boldsymbol{M}_t = a_t \boldsymbol{M}_{t-1} + \boldsymbol{k}_t^T \boldsymbol{v}_t$ |
| RWKV7 | $\boldsymbol{M}_t = (\boldsymbol{a}_t^T \mathbf{1}) \boldsymbol{M}_{t-1} + b_t \nabla l(\boldsymbol{M}_{t-1}; \boldsymbol{k}_t, \boldsymbol{v}_t)$ |

memory by weighted summation, the query can effectively retrieve information from this mixed memory, and generate attention output followed by other layers.

**Shared Memory.** To enhance our model's ability to capture long-term dependencies, we introduce a *shared memory* mechanism. This shared memory has access to the entire sequence information, allowing it to effectively store and retrieve long-term information. By integrating shared memory into our model, we ensure that it can leverage the complete historical context, resulting in significant improvements in performance and robustness.

### 3.3 HARDWARE-EFFICIENT IMPLEMENTATION

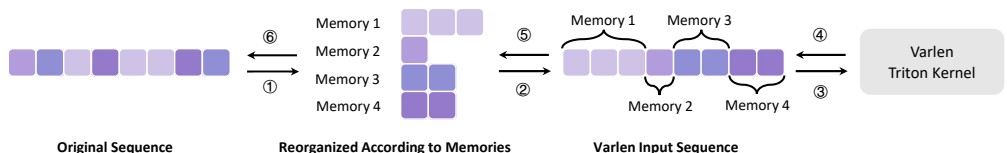

Figure 2: **Hardware-efficient Implementation of MoM.** Tokens sharing the same color are routed to the same memory. ① Tokens are first split into groups according to memory routing results, ② then concatenated into a varlen input sequence, ③ processed by the Triton kernel, ④ the outputs are returned, ⑤ split back into their respective memories, and ⑥ finally restored to the original sequence order. For clarity, the illustration shows the top-1 routing case, and the $qkv$ projection is omitted.

In the implementation of MoM, mixing memories before query multiplication is equivalent to multiplying each memory by the query and then mixing the results, allowing us to reuse efficient Triton-based operators from prior linear sequence models. We first reorder the sequence tokens according to the routing results so that they follow the memory layout. The reordered tokens are then concatenated with varlen for operator computation, after which the results are aggregated via weighted summation. In this way, MoM's computation can be effectively reduced to **varlen operations**, enabling efficient execution. We elaborate on this process below.

Given input tokens $\boldsymbol{x}_{b,t} \in \mathbb{R}^d$ for batch $b \in \{1, \ldots, B\}$ and time step $t \in \{1, \ldots, T\}$, each token is routed to one or more memories $m \in \{1, \ldots, M\}$ with routing weights $\alpha_{b,t,m} \geq 0$ satisfying $\sum_{m=1}^{M} \alpha_{b,t,m} = 1$.

For each $(b, m)$, define the ordered index set

$$\mathcal{I}_{b,m} = \big(t_{b,m}(1), \ldots, t_{b,m}(L_{b,m})\big),$$

where $t_{b,m}(j)$ is the original sequence index of the $j$-th token assigned to memory $m$, and $L_{b,m} = |\mathcal{I}_{b,m}|$. We index buckets lexicographically by $p = (b-1)M + m$ and define cumulative boundaries

$$s_0 = 0, \qquad s_p = \sum_{q=1}^{p} L_q \quad (p = 1, \ldots, BM).$$

The flattened sequence $\tilde{\boldsymbol{X}}$ is obtained by

$$\tilde{\boldsymbol{x}}_{s_{p-1}+j} = \boldsymbol{x}_{b,\, t_{b,m}(j)}, \qquad j = 1, \ldots, L_{b,m},$$

with varlen representation $(\tilde{\boldsymbol{X}}, \boldsymbol{s})$, where $\boldsymbol{s} = (s_0, \ldots, s_{BM})$.

For each bucket $p = (b-1)M + m$, queries share a projection matrix $\boldsymbol{W}_Q$, while keys and values use memory-specific projections $\boldsymbol{W}_K^{(m)}, \boldsymbol{W}_V^{(m)}$:

$$\tilde{\boldsymbol{q}}_u = \boldsymbol{W}_Q \tilde{\boldsymbol{x}}_u, \quad \tilde{\boldsymbol{k}}_u = \boldsymbol{W}_K^{(m)} \tilde{\boldsymbol{x}}_u, \quad \tilde{\boldsymbol{v}}_u = \boldsymbol{W}_V^{(m)} \tilde{\boldsymbol{x}}_u, \quad u \in \{s_{p-1}+1, \ldots, s_p\}.$$

A memory-specific kernel $\mathcal{F}_m$ with parameters $\boldsymbol{\theta}^{(m)}$ is applied independently to each segment:

$$\boldsymbol{o}_{s_{p-1}+1:s_p} = \mathcal{F}_m\big(\tilde{\boldsymbol{q}}_{s_{p-1}+1:s_p}, \tilde{\boldsymbol{k}}_{s_{p-1}+1:s_p}, \tilde{\boldsymbol{v}}_{s_{p-1}+1:s_p}; \boldsymbol{\theta}^{(m)}\big).$$

Mapping outputs back to the original sequence, the $j$-th token in $\mathcal{I}_{b,m}$ has per-memory output

$$\hat{\boldsymbol{o}}_{b,t_{b,m}(j),m} = \boldsymbol{o}_{s_{p-1}+j}.$$

Finally, token-level representations are reconstructed by weighted summation:

$$\boldsymbol{y}_{b,t} = \sum_{m=1}^{M} \alpha_{b,t,m} \, \hat{\boldsymbol{o}}_{b,t,m}.$$

## 4 EXPERIMENTS

### 4.1 EXPERIMENTAL SETUPS

**Models.**  In our experiments, we employ the Gated DeltaNet (Yang et al., 2024) as the memory update mechanism in MoM. The model is configured with four memory states, two of which are activated at each time step, along with a shared memory.

**Baselines.**  We evaluate MoM against several linear recurrent models and Transformers, including RetNet (Sun et al., 2023), GLA (Yang et al., 2023), Gated DeltaNet (Yang et al., 2024), and Transformer++ (Touvron et al., 2023), which incorporates Rotary Position Embeddings (Su et al., 2024) and GLU (Shazeer, 2020) into the Transformer architecture. To ensure a fair comparison, we train all baseline models from scratch using the exact same number of tokens.

**Training.**  We follow the training procedure described by Yang et al. (2023), utilizing the SlimPajama dataset (Soboleva et al., 2023) sampled with 100B tokens and tokenized using the Mistral tokenizer (Jiang et al., 2023). We train models from scratch with parameter sizes of 380M and 1.3B, respectively. For the 380M models, we train on 15B tokens with a batch size of 0.5M tokens. More detailed training configuration is provided in Appendix C. We utilized publicly available pretrained weights from Zhang et al. (2024) with exactly same configuration [1].

**Parameter Explanation.**  We report model sizes using the common shorthand, where "380M" denotes a configuration with 24 layers and hidden size 1024, and "1.3B" denotes 24 layers with hidden size 2048. The main goal of MoM is to expand the memory capacity of linear sequence models through sparse activation. To this end, we apply sparse activation only to the key and value projections, which results in a small increase in activated parameters that is well justified by the performance gains. A detailed discussion on fairness is provided in Appendix G.

### 4.2 MAIN RESULTS

#### 4.2.1 RECALL-INTENSIVE TASKS

Linear sequence models, due to their limited memory capacity, often exhibit a significant performance gap compared to Transformer models, especially in recall-intensive tasks where extensive context is crucial. These tasks highlight notable performance differences among various linear models, making them a more accurate benchmark for evaluating a linear model's capabilities in handling contextual information.

To thoroughly assess our model's proficiency in such scenarios, we test six recall-intensive tasks following Arora et al. (2024): FDA (Arora et al., 2023), SWDE (Arora et al., 2023; Lockard et al., 2019), SQuAD (Rajpurkar et al., 2018), NQ (Kwiatkowski et al., 2019), TriviaQA (Joshi et al., 2017) and Drop (Dua et al., 2019). These tasks are designed to challenge a model's ability to perform context-based retrieval and comprehension.

As shown in Table 2, our proposed approach, benefiting from increased memory capacity and memory mixing mechanism, achieves significant improvements over other linear sequence models. Specifically, our model effectively narrows the performance gap with Transformer models. This improvement underscores the advantage of our method in capturing and utilizing long-range dependencies, thereby enhancing performance on tasks that require extensive contextual understanding.

#### 4.2.2 LONG CONTEXT TASKS

---

[1]Models marked with an asterisk $^{\dagger}$ use open-source pretrained weights with identical training configurations.

Table 2: **Results on Recall-Intensive Tasks.** All inputs are truncated to a maximum length of 2K tokens. MoM significantly outperforms all other linear models across both model sizes. In the 1.3B model, MoM even achieves performance very close to that of Transformer models.

| Scale | Model | FDA | SWDE | SQUAD | NQ | TriviaQA | Drop | Avg. | Avg. (no FDA) |
|---|---|---|---|---|---|---|---|---|---|
| *380M Params* | Transformer++ | 46.14 | 25.87 | 33.22 | 18.94 | 45.97 | 20.03 | 31.70 | 28.81 |
| *15B Tokens* | RetNet | 5.90 | 9.28 | 22.41 | 6.91 | 40.05 | 18.59 | 17.19 | 19.45 |
| *L=24, d=1024* | HGRN2 | 11.53 | 17.34 | 24.08 | 12.67 | 43.84 | 17.35 | 21.14 | 23.06 |
| | GLA | 11.26 | 16.78 | 27.85 | 12.77 | 43.90 | 17.68 | 21.71 | 23.80 |
| | GSA | 6.36 | 16.87 | 21.90 | 14.60 | 42.18 | 16.72 | 19.77 | 22.45 |
| | Gated DeltaNet | 20.53 | 23.24 | 28.55 | 14.98 | 44.91 | 16.48 | 24.78 | 25.63 |
| | MoM | **22.98** | **29.90** | **29.69** | **16.60** | **48.82** | **20.99** | **28.16** | **29.20** |
| *1.3B Params* | Transformer++[†] | 44.32 | 32.43 | 42.59 | 24.49 | 58.47 | 21.56 | 37.31 | 35.91 |
| *100B Tokens* | RetNet[†] | 13.62 | 22.59 | 33.46 | 15.43 | 53.79 | 19.79 | 26.45 | 29.01 |
| *L=24, d=2048* | HGRN2[†] | 12.35 | 23.24 | 33.19 | 19.10 | 55.27 | 19.65 | 27.13 | 30.09 |
| | GLA[†] | 27.61 | 30.93 | 35.04 | 22.27 | 56.28 | 19.45 | 31.93 | 32.79 |
| | GSA[†] | 23.25 | 32.80 | 35.57 | 22.96 | 57.05 | 20.65 | 32.05 | 33.81 |
| | Gated DeltaNet | 30.25 | 27.65 | 34.06 | 23.22 | 58.23 | 20.36 | 32.30 | 32.70 |
| | MoM | **41.14** | **34.30** | **37.08** | **24.11** | **58.59** | **21.03** | **36.04** | **35.02** |

Assessing performance on long-context tasks is crucial for linear models, as it reflects their ability to handle long-range dependencies effectively. We evaluated our model's comprehension of long contexts using the Long-Bench benchmark (Bai et al., 2024; Contributors, 2023). In Table 3, we present the average results across various categories, including summarization, few-shot learning, synthetic tasks, and code completion, along with the overall mean across all tasks. The complete detailed results are provided in Appendix I.

Table 3: **LongBench Benchmark Results.** *Note:* Sum = Summarization, FS = Few-shot, Syn = Synthetic.

| Model | Sum | FS | Syn | Code | Avg. |
|---|---|---|---|---|---|
| RetNet[†] | 6.30 | 15.76 | **2.64** | 40.52 | 13.61 |
| HGRN2[†] | 6.51 | 15.50 | 2.61 | 40.11 | 13.02 |
| GSA[†] | **7.75** | 20.29 | 1.92 | 42.83 | 14.61 |
| Gated DeltaNet | 7.14 | 18.00 | 2.10 | 41.52 | 13.98 |
| MoM | 6.89 | **21.26** | 2.63 | **47.79** | **15.64** |

Table 4: **Comparison of Mixture of Memories and Single Memory Expanded.** We constructed MoM models using different memory update mechanisms. Separate memory segments yielded better performance compared to simply increasing the memory capacity of a single memory.

| Model | Params | ARC-e acc↑ | ARC-c acc_n↑ | Hella. acc_n↑ | Lamb. acc↑ | PIQA acc↑ | Wino. acc↑ | Avg. |
|---|---|---|---|---|---|---|---|---|
| GLA *expanded* | 425M | 42.34 | 22.95 | 34.56 | 20.45 | 63.00 | **50.12** | 38.90 |
| GLA *MoM* | 395M | **42.85** | **24.15** | **36.60** | **23.23** | **63.22** | 49.88 | **39.99** |
| Gated DeltaNet *expanded* | 550M | 43.60 | 24.66 | **37.80** | 26.90 | 64.47 | 50.51 | 41.32 |
| Gated DeltaNet *MoM* | 444M | **44.65** | **24.74** | 36.54 | **27.93** | **66.16** | **51.78** | **41.97** |

| Model | Params | FDA | SWDE | SQUAD | NQ | TriviaQA | Drop | Avg. |
|---|---|---|---|---|---|---|---|---|
| GLA *expanded* | 425M | **15.08** | 20.15 | 28.28 | 13.30 | 41.65 | 18.74 | 22.87 |
| GLA *MoM* | 395M | 9.90 | **21.65** | **29.36** | **14.16** | **45.20** | **20.89** | **23.53** |
| Gated DeltaNet *expanded* | 550M | 18.26 | 24.27 | **30.03** | **17.74** | 48.34 | 19.26 | 26.32 |
| Gated DeltaNet *MoM* | 444M | **22.98** | **29.90** | 29.69 | 16.60 | **48.82** | **20.99** | **28.16** |

### 4.2.3 MIXED MEMORY VS. SINGLE MEMORY

To validate the effectiveness of our mixed memory mechanism, we compare our MoM model with mixed memories to a baseline model that uses an expanded single memory with the same activated memory capacity. We adopt the same memory update method as existing linear models and extend it within our MoM framework. For comparison, we employed the commonly used method of expanding the single memory by expanding the dimension of $v$ to match the total size of all activated memories in the MoM model. We evaluate their performance on common-sense reasoning tasks and recall-intensive tasks in Table 4.

Table 5: **Comparison with the Same Activated Parameters.** MoM and Gated DeltaNet with 400M activated parameters are tested.

| Model | Params | ARC-e acc↑ | ARC-c acc$_n$↑ | Hella. acc$_n$↑ | Lamb. acc↑ | PIQA acc↑ | Wino. acc↑ | Avg. |
|---|---|---|---|---|---|---|---|---|
| Gated DeltaNet | 400M | 46.04 | 23.55 | 35.18 | **27.01** | 66.05 | 50.83 | 41.44 |
| MoM | 400M | **47.10** | **23.72** | **35.43** | 26.88 | 64.64 | **51.22** | **41.50** |

| Model | Params | FDA | SWDE | SQUAD | NQ | TriviaQA | Drop | Avg. |
|---|---|---|---|---|---|---|---|---|
| Gated DeltaNet | 400M | 20.53 | 23.24 | 28.55 | 14.98 | 44.91 | 16.48 | 24.78 |
| MoM | 400M | **24.16** | **25.59** | **29.46** | **15.36** | **46.15** | **18.35** | **26.51** |

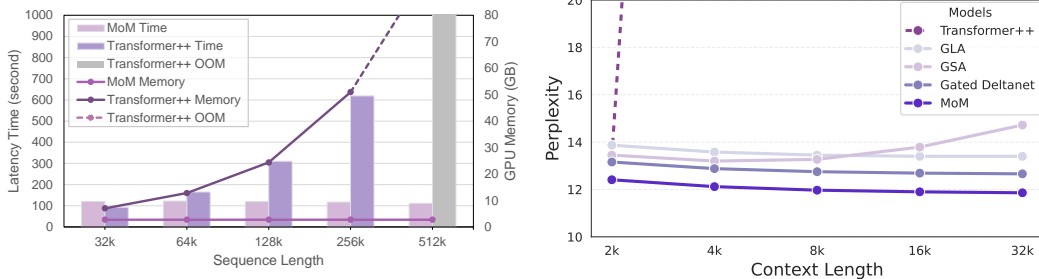

Figure 3: **Inference Efficiency of MoM.** We demonstrate the inference time and GPU memory consumption required to generate 1K tokens at specific sequence lengths.

Figure 4: **Length Extrapolation.** We extrapolated models trained on 2K sequences to a length of 32K for perplexity (ppl) evaluation.

The experimental results demonstrated that using multiple mixed memories leads to a greater improvement than simply expanding the capacity of a single memory with less parameters. This confirms that mixed memory can effectively reduce interference from different inputs. Assigning inputs specifically to different memories, combined with the use of a forget gate, proves to be a more effective approach for reducing interference than relying solely on a forget gate.

### 4.2.4 EFFICIENCY

We compare the inference speed and memory usage of MoM and Transformer++ with flash attention in Fig 3. Our analysis demonstrates that MoM exhibits linear complexity, showcasing significant advantages over the Transformer model when handling long sequences. Specifically, MoM's efficient memory update mechanisms allow it to process longer inputs with reduced computational overhead, positioning it as a more scalable solution for large-scale natural language processing tasks.

### 4.2.5 LENGTH EXTRAPOLATION

We pretrained the models on the Slimpajama dataset with a 2K context length and conducted extrapolation experiments on various lengths using the Fineweb (Penedo et al., 2024) dataset. We extended the length to 32K to calculate perplexity (ppl). As shown in Fig 4, the Transformer model experienced a significant increase in ppl due to its poor extrapolation capability. Among the linear models, MoM achieved the best results.

### 4.2.6 MEMORY ANALYSIS

**Memory Load Balance Analysis.** To evaluate whether each memory segment in MoM is effectively balanced during inference on downstream tasks, we analyzed the number of tokens routed to each layer using around 300k tokens from the ARC-easy benchmark. We visualized the results with auxiliary loss (following the formulation introduced in Switch Transformer (Fedus et al., 2022)) in Fig 6 with heatmaps and we also visualized results with auxiliary loss in Fig 10. Due to the adoption of auxiliary loss, the memory segments in each layer are almost uniformly routed and activated.

**Memory Specialization Analysis.** To quantitatively investigate whether the router guides memories toward specialized roles, we analyzed the routing decisions within the model's deep layers during inference on the ARC-easy benchmark. We sampled the input hidden states ($x_t$) for a large number of tokens and used UMAP (McInnes et al., 2018) to project these high-dimensional states into a 2D space. Each point in the visualization is color-coded by its Top-1 routed memory index.

The results are presented in Figure 5. We observe a **clear and distinct clustering phenomenon**, which confirms that the router has learned a meaningful specialization. Since our model employs top-$k$=2 routing, each token is routed to two memories. For visualization clarity, the figure only plots the top-1 memory destination. Consequently, some overlap at the cluster boundaries is visible, which is an expected outcome of this design.

This clustering analysis also provides a new perspective for understanding MoM from a Test-Time Training (TTT) point of view. The memory update mechanism we use, Gated DeltaNet, adopts a Delta Rule learning style, dynamically fitting a $k \to v$ mapping at test time (with the optimization goal $kM = v$). When test data is highly discrete or widely distributed in the feature space, a single memory network $M$ struggles to fit all the data quickly and accurately. Our UMAP analysis demonstrates that the MoM router acts as a dynamic clustering mechanism, automatically partitioning the broad input data stream during inference into multiple, more concentrated, and cohesive subsets. It then assigns each subset to a specialized memory for processing. This is equivalent to implementing a form of **TTT Ensemble Learning**: each memory $M_m$ no longer needs to fit the entire complex data distribution, but only a simpler sub-distribution, thereby reducing the learning difficulty.

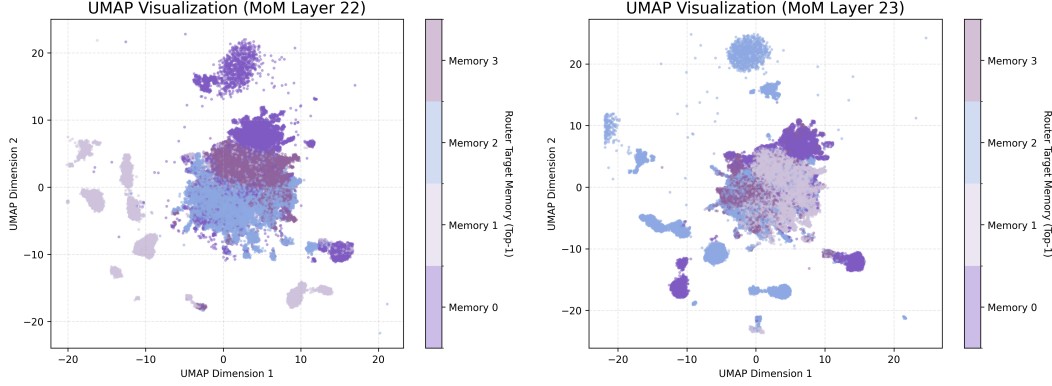

Figure 5: **UMAP visualization of memory specialization.** Token hidden states are colored by their top-1 memory destination. The distinct clustering suggests a learned specialization. The partial overlap is an expected artifact of plotting the top-1 destination for a top-$k$=2 router.

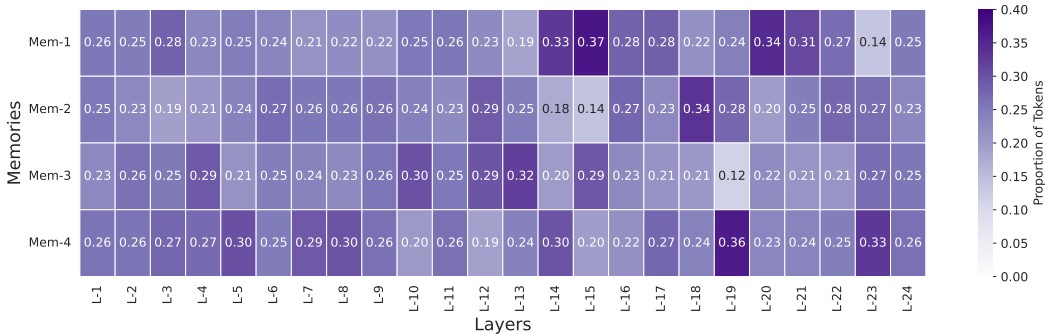

Figure 6: **Memory Load Balance Analysis.** Token Routing Distribution Across Layers and Memories with Aux Loss.

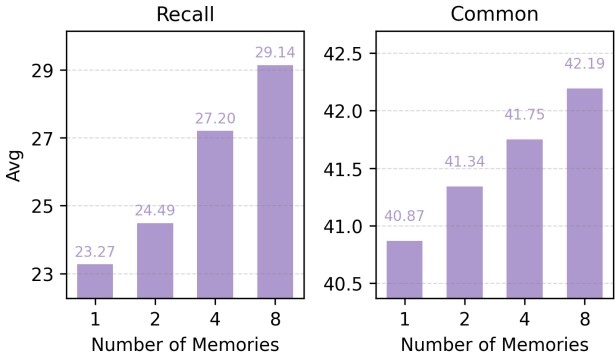

Figure 7: Scaling performance with increasing number of memories with a fixed activation ratio of 0.5.

Table 6: Ablation on memory count and shared memory, showing average accuracy across recall-intensive tasks.

|  | Recall ↑ | Common ↑ |
|---|---|---|
| **Aux Loss Scale** | | |
| 1e-2 | 27.59 | **42.10** |
| 5e-3 | 26.55 | 41.71 |
| 1e-3 | **28.16** | 41.97 |
| 0 | 27.23 | 41.58 |
| **Shared Memory** | | |
| w/ shared memory | **28.16** | 41.97 |
| w/o shared memory | 26.06 | 40.38 |

### 4.2.7 MoM Scaling Up & Ablation Study

We examine the effect of scaling both the number of memory states and the number of top-$k$ activations in MoM. To ensure comparability, Fig. 7 reports results with a fixed activation ratio of 0.5. Increasing the number of memories from 1 to 8 consistently improves performance across both recall-intensive and commonsense benchmarks. These results indicate that enlarging the memory pool effectively mitigates interference and enhances capacity. More comprehensive results covering other activation ratios and activation settings are provided in Appendix H.

We further study the influence of auxiliary loss and shared memory in MoM, using a 380M-parameter model trained on 15B tokens. As shown in Table 6, auxiliary loss improves stability and performance when applied with a suitable weight. In addition, shared memory consistently benefits performance with global information. These results highlight the complementary roles of auxiliary loss and shared memory in stabilizing and enhancing MoM.

## 5 CONCLUSION

In this paper, we propose Mixture-of-Memories (MoM), a novel architecture that enhances memory capacity and eliminates memory interference. By leveraging multiple independent memory states, MoM significantly improves performance on recall-intensive tasks while maintaining the efficiency advantages of linear models. Instead of simply discarding tokens as done in gating mechanisms, our memory separation paradigm provides a more effective way to preserve sequence information. Our experimental results demonstrate that MoM outperforms existing linear sequence modeling methods, particularly on tasks requiring strong recall, and achieves performance comparable to Transformer models. This makes MoM a promising approach for applications need strong efficiency and recall-intensive performance, paving the way for efficient sequence modeling.

## 6 ETHICS STATEMENT

This work does not involve human subjects, sensitive data, or high-risk applications. All experiments are conducted on publicly available datasets. We encourage responsible and ethical use of the proposed methods in line with community standards.

## 7 REPRODUCIBILITY STATEMENT

Our code is released at https://github.com/OpenSparseLLMs/MoM and all the models we trained from scratch will be released at huggingface. We provide the training and evaluation scripts to make sure that all the results in the paper can be easily reproduced.

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

## A    RELATED WORK

### LINEAR RECURRENT MODELS

Linear recurrent models, comprising linear attention, linear RNNs, state-space models (SSMs), have garnered significant research interests (Qin et al., 2023b). The advancement of SSMs began with the pioneering work on S4 (Gu et al., 2022), which was later optimized through a diagonalized version (Gupta et al., 2022). Despite their strong performance on the LRA benchmark, these models have faced challenges in language modeling mainly because they rely solely on data-independent processes. As research progressed, constant forgetting gates were introduced, helping to alleviate some interference issues by uniformly managing memory decay (Sun et al., 2023; Gu & Dao, 2024). The next breakthrough involved data-dependent forget gates. These allowed models to dynamically adjust memory updates based on the input data, significantly enhancing performance across various tasks (Qin et al., 2024c;d; Yang et al., 2023; Zhang et al., 2024; Yang et al., 2024; Qin et al., 2024a). Sequence parallelism techniques are well adapted on linear recurrent models (Sun et al., 2024a; 2025) for efficient long context training. There have also been recent advancements in scaling law and test-time regression optimization (Shen et al., 2024; Sun et al., 2024c; Behrouz et al., 2024).

Building on these advancements, our MoM model incorporates data-dependent mechanisms that selectively update memory. By efficiently managing interference through tailored memory updates and leveraging increased memory capacity, MoM represents a further evolution, improving model expressiveness and performance.

### MIXTURE-OF-EXPERTS

Mixture-of-Experts (MoE) is a technique designed to enhance the capacity of deep neural networks while maintaining computational efficiency (Fedus et al., 2022; Rajbhandari et al., 2020; Lepikhin et al., 2020; Tang et al., 2023; Zhu et al., 2024; Qu et al., 2024). MoE achieves this by activating a subset of parameters, known as "experts", for each input, which reduces the computational costs. Shazeer first integrated MoE into LSTM layers (Shazeer et al., 2017). The Switch Transformer (Fedus et al., 2022) refined this approach by simplifying the gating mechanism to select only one expert per input. Gshard (Lepikhin et al., 2020) further advanced this by using a top-2 expert routing strategy to improve performance. Recent MoE models, such as Deepseek-MoE (Dai et al., 2024), introduce shared experts to capture and consolidate common knowledge across different contexts, while designing fine-grained experts to increase combinatorial flexibility.

## B    COMPARISON BETWEEN MoM AND MoE

While our approach to implementing the Mixture-of-Memories (MoM) draws inspiration from the Mixture-of-Experts (MoE) framework, there are notable differences that distinguish our method from traditional MoE implementations.

- **Purpose**: The MoE was introduced to scale up the number of parameters without significantly increasing computational resources. It address the limitations of dense models in scaling both parameters and computational demands through sparse activation. However, MoM is designed to expand the memory capacity of linear attention models while preserving their linear time complexity. By sparsely activating memories and using weighed summation to create a mixed memory, MoM effectively address the challenge of forgetting historical information in linear attention. Moreover, by separating the memory into distinct states, MoM reduces interference between different pieces of information.

- **Structure**: In conventional MoE, each expert is a separate neural network within the feedforward network (FFN) layer such as Qwen-MoE (Team, 2024) and Linear-MoE Sun et al. (2024a). In contrast, in MoM, each memory is an RNN state with distinct key-value projection weights to generate different key-value pairs. MoE operates during the channel mixing phase, where each token is processed independently by selected experts. On the other hand, MoM functions during the token mixing phase, where each memory processes different segments of the sequence, preserving inter-token relationships.

Table 7: **Results on Common-Sense Reasoning Tasks.** The performance of linear models and Transformer models is comparable; however, MoM consistently achieves the best average performance across all model sizes.

| Scale | Model | Wiki. ppl↓ | Lamb. ppl↓ | ARC-e acc↑ | ARC-c acc$_n$↑ | Hella. acc$_n$↑ | Lamb. acc↑ | PIQA acc↑ | Wino. acc↑ | Avg. |
|---|---|---|---|---|---|---|---|---|---|---|
| *380M Params* | Transformer++ | 26.88 | 76.46 | 44.91 | **25.94** | 34.95 | 26.90 | 64.31 | 51.07 | 41.35 |
| *15B Tokens* | RetNet | 31.07 | 87.11 | 44.49 | 23.04 | 33.86 | 23.93 | 63.49 | 52.33 | 40.19 |
| *L=24, d=1024* | HGRN2 | 27.90 | 77.40 | 45.24 | 23.63 | 35.61 | 24.74 | 65.45 | **54.06** | 41.46 |
| | GLA | 28.78 | 79.95 | 44.53 | 22.27 | 34.84 | 24.94 | 63.93 | 51.38 | 40.32 |
| | GSA | 28.17 | 82.50 | 45.50 | 24.23 | 35.00 | 24.02 | 64.85 | 50.43 | 40.67 |
| | Gated DeltaNet | 26.47 | 58.59 | **46.04** | 23.55 | 35.18 | 27.01 | 66.05 | 50.83 | 41.44 |
| | MoM | **25.86** | **55.41** | 44.65 | 24.74 | **36.54** | 27.93 | **66.16** | 51.78 | **41.97** |
| *1.3B Params* | Transformer++[†] | 17.61 | 19.29 | 55.01 | 28.07 | 49.21 | 40.95 | 70.08 | 56.27 | 49.93 |
| *100B Tokens* | RetNet[†] | 18.18 | 21.97 | 57.49 | 26.88 | 48.09 | 37.75 | 69.37 | 53.28 | 48.81 |
| *L=24, d=2048* | HGRN2[†] | 17.32 | 15.65 | **58.33** | 28.07 | **51.93** | 42.31 | 71.33 | 52.01 | 50.66 |
| | GLA[†] | 17.61 | 19.66 | 55.18 | 27.56 | 48.89 | 40.03 | 69.86 | 53.91 | 49.24 |
| | GSA[†] | 16.69 | 16.02 | **58.33** | **28.33** | 50.98 | 42.03 | **72.25** | 53.43 | 50.89 |
| | Gated DeltaNet | 17.14 | 18.80 | 56.82 | 27.39 | 49.77 | 39.94 | 71.76 | 51.78 | 49.58 |
| | MoM | **16.64** | **14.83** | 55.35 | 27.99 | 50.95 | **43.43** | 71.27 | **56.83** | **50.97** |

## C  EXPERIMENTS DETAILS

For the 380M models, we train on 15B tokens with a batch size of 0.5M tokens. The warmup tokens count is set to 0.25M. We set the hidden ratio of our model to 3 to keep the activated parameter count approximately the same. For the 1.3B models, we train on 100B tokens with a batch size of 2M tokens. The warmup tokens count is 1B. We employ AdamW optimizer (Loshchilov et al., 2017; Sun et al., 2024b) with learning rate of 3e-4 with cosine learning rate schedule (Zhou et al., 2020). The weight decay is set to 0.01 and gradient clipping is 1.0. Our experiments were conducted using 32 NVIDIA A800 GPUs. Training the 380M parameter model required approximately 10 hours, while the 1.3B parameter model took around 6 days.

## D  COMMONSENSE REASONING TASKS

As shown in Table 7, we report the language modeling perplexity and zero-shot performance of commonsense reasoning tasks following (Zhang et al., 2024) which includes WikiText (Merity et al., 2016), LAMBADA (Paperno et al., 2016), ARC-easy, ARC-challenge (Clark et al., 2018), HellaSwag (Zellers et al., 2019), PiQA (Bisk et al., 2020) and WinoGrande (Sakaguchi et al., 2019). The evaluation results are based on the lm-evaluation-harness (Gao et al., 2024).

Experimental results show that MoM outperforms other linear models and surpassed the Transformer model as well.

## E  TRAINING LOSS COMPARISON

To further assess the learning efficiency of MoM, we compared the training loss curves of MoM with those of other baseline models. As depicted in Figure 8, MoM consistently maintains the lowest loss throughout the entire training phase. Even as training nears convergence, MoM continues to exhibit a clear advantage over other methods.

## F  THE HYBRID OF MOM AND TRANSFORMER

We delve deeper into the hybridization of MoM and Transformer layers by integrating 1 Transformer layer after every 7 MoM layers, resulting in only 3 Transformer layers across a total of 24 layers. The performance on commonsense reasoning and recall-intensive tasks is presented in the table 8. MoM-Hybrid demonstrates significantly improved results compared to Transformer models, despite using only 3 layers of global attention.

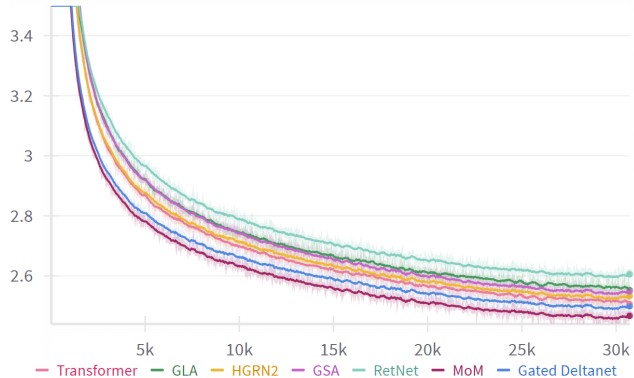

Figure 8: **Training Loss.** Loss curves for training 380M models on 15B tokens with a fixed random seed of 42.

Table 8: **Hybrid Model Performance.** The hybrid model integrates 1 Transformer layer after every 7 MoM layers, resulting in only 3 Transformer layers across a total of 24 layers.

| Model | FDA | SWDE | SQUAD | NQ | TriviaQA | Drop | Avg. |
|---|---|---|---|---|---|---|---|
| Transformer++ | 46.14 | 25.87 | 33.22 | 18.94 | 45.97 | 20.03 | 31.70 |
| MoM | 22.98 | 29.90 | 29.69 | 16.60 | **48.82** | **20.99** | 28.16 |
| MoM *Hybrid* | **58.13** | **44.05** | **35.71** | **20.18** | 48.10 | 20.60 | **37.80** |

| Model | ARC-e acc↑ | ARC-c $acc_n$↑ | Hella. $acc_n$↑ | Lamb. acc↑ | PIQA acc↑ | Wino. acc↑ | Avg. |
|---|---|---|---|---|---|---|---|
| Transformer++ | 44.91 | **25.94** | 34.95 | 26.90 | 64.31 | 51.07 | 41.35 |
| MoM | 44.65 | 24.74 | **36.54** | 27.93 | **66.16** | 51.78 | 41.97 |
| MoM *Hybrid* | **46.55** | 24.49 | 36.45 | **28.86** | 65.51 | **52.41** | **42.38** |

## G FAIRNESS

To enhance memory capacity, MoM applies sparse activation to the key and value projections. Although these projections constitute a small portion of the overall model parameters, this inevitably increases the parameter count. Due to differing linear model structures, aligning both parameter count and memory capacity exactly is challenging. Thus, to ensure fairness, we conduct comparisons from two perspectives: **equal activated parameter count** and **equal memory capacity**.

### G.1 EQUAL ACTIVATED PARAMETER COUNT

To ensure a fair comparison of parameters, we reduced the MLP hidden ratio to 2 and retrained the MoM model using the same training configurations as in Section 4.1. Both MoM and Gated Deltanet were set with 400M activated parameters. Although the smaller hidden ratio might impact the model's commonsense knowledge, we tested on commonsense reasoning tasks and recall-intensive benchmarks. MoM consistently outperformed Gated Deltanet in both tests, further validating the effectiveness of the MoM approach. The results are presented in Table 5

### G.2 EQUAL MEMORY CAPACITY

To ensure a fair comparison of memory capacity, we also compared the single extended memory model with the MoM model. Notably, the single extended memory model has more parameters than the activated parameters in the MoM due to the extension of the $v$ dimension. MoM expands memory more elegantly and significantly outperforms in both recall and commonsense tasks. This comparison result is presented in Table 4.

## H    DETAILED SCALING RESULTS

To further examine the scalability of MoM from the perspective of memory capacity, we evaluate the effect of enlarging the memory pool beyond the main settings. Specifically, we compare two activation ratios, where the number of active memories accounts for either 0.5 or 0.25 of the total memory states. In all cases, an additional shared memory is included. Starting from a single memory as the baseline, we expand the number of memories to 2, 4, 8, and 16, and report the averaged results on both recall-intensive and commonsense benchmarks. The results, shown in Fig. 9, indicate that under a fixed activation ratio, increasing the memory size consistently improves performance on both categories of tasks.

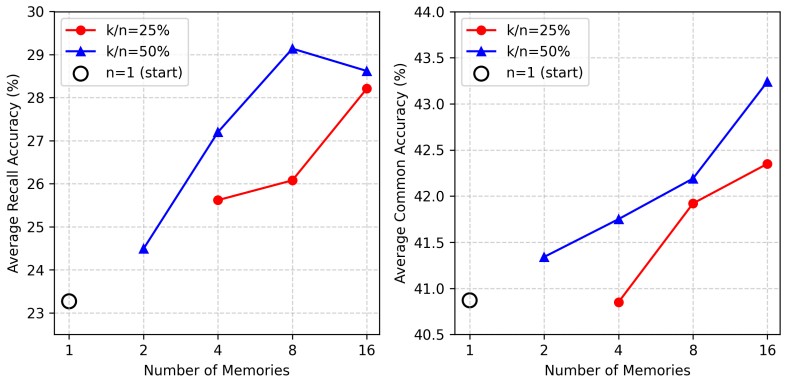

Figure 9: **Detailed scaling results.** Performance shows a general improvement on both recall-intensive and commonsense tasks as the number of memories increases.

## I    FULL RESULTS

### I.1    ABLATION ANALYSIS ON SHARED AND SPLIT MEMORIES

To rigorously verify the necessity of the mixture mechanism, we further disentangle the contributions of the shared memory and the split memories in Table 9. As discussed in Section 3.2.2, the shared memory is designed to capture global context, serving as a complementary component to the split memories which specialize in local token subsets. It is important to note that the "Shared Memory Only" configuration is mathematically equivalent to the standard Gated DeltaNet baseline.

Table 9: **Ablation study disentangling the effects of shared and split memories.**

| Model Configuration | Avg. (Recall) |
| --- | --- |
| Shared Memory Only | 24.78 |
| Split Memory Only | 26.06 |
| **Mixed Memory (MoM)** | **28.16** |

### I.2    ROBUSTNESS TO ROUTER INITIALIZATION

To verify the stability of the routing mechanism, we conducted an ablation study on router initialization. Using the 380M model trained on 5B tokens, we fixed the initialization seed for all other model parameters (seed=42) while varying the random seed specifically for the router network.

As shown in Table 10, the variations in both validation loss and downstream task performance (average accuracy on commonsense benchmarks) are negligible, with the loss variance remaining within 0.07%. This empirical evidence confirms that the MoM architecture is robust to router initialization strategies.

Table 10: Sensitivity analysis of router initialization. Models were trained for 5B tokens.

| Router Seed | Loss | Avg. (Commonsense) |
|---|---|---|
| 1 | 2.9396 | 40.37 |
| 42 | 2.9409 | 40.20 |
| 1234 | 2.9426 | 40.07 |

| Model | SQA | MQA | Sum | FS | Syn | Code | Zh-Avg | En-Avg | Avg. |
|---|---|---|---|---|---|---|---|---|---|
| RetNet | **9.23** | **7.23** | 6.3 | 15.76 | **2.64** | 40.52 | 15.44 | 13.5 | 13.61 |
| HGRN2 | 7.38 | 6.02 | 6.51 | 15.5 | 2.61 | 40.11 | 14.28 | 13.12 | 13.02 |
| GSA | 8.21 | 6.63 | **7.75** | 20.29 | 1.92 | 42.83 | 15.06 | 15.2 | 14.61 |
| Gated DeltaNet | 8.52 | 6.61 | 7.14 | 18 | 2.1 | 41.52 | 14.19 | 14.63 | 13.98 |
| MoM | 8.14 | 7.11 | 6.89 | **21.26** | 2.63 | **47.79** | **17.33** | **15.71** | **15.64** |

Table 11: **Complete Results of LongBench.** SQA: Single-doc QA, MQA: Multi-doc QA, Sum: Summarization, FS: Few-shot learning, Syn: Synthetic

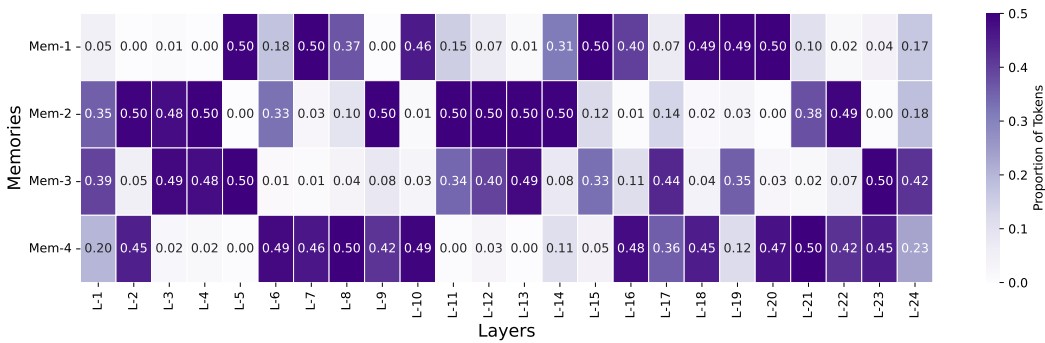

Figure 10: **Memory Load Balance Analysis.** Token Routing Distribution Across Layers and Memories without Aux Loss.

## J  THE USE OF LLMS

Large language models (LLMs) were only used to refine the grammar and spelling of some paragraphs in this paper. They were not used for generating research ideas, designing experiments, or writing substantive content.

