# OpenReview forum: "MoM: Linear Sequence Modeling with Mixture-of-Memories"
_ICLR.cc/2026/Conference — ICLR 2026 Poster_

### Official Review · Reviewer_51kq · 2025-10-27

**Soundness:** 3
**Presentation:** 3
**Contribution:** 3
**Rating:** 6
**Confidence:** 3

**Summary:**

This paper proposes Mixture-of-Memories (MoM), a new architecture designed to improve the performance of linear sequence modeling methods such as linear attention, state space models, and linear RNNs. The key idea is to maintain multiple independent memory states instead of a single fixed-size state, with a router network assigning input tokens to specific memory slots. This design aims to enhance memory capacity and reduce interference between token representations, addressing a common limitation in existing linear models when handling recall-intensive tasks. MoM is presented as a general and flexible framework that can be integrated with various linear memory update mechanisms. The authors report that MoM achieves better results than prior linear sequence models across multiple language tasks—particularly those requiring long-term recall—and approaches the performance of Transformer-based architectures, while preserving linear training complexity and constant inference cost.

**Strengths:**

1.	The paper is well-written, with a clearly motivated method and strong experimental evidence to support its claims.

**Weaknesses:**

1.	Experiments with larger model sizes could further strengthen the paper.

2.	Some hyperparameter choices—such as the top-k value in the routing process and the predefined number of memory slots—are not thoroughly discussed.

**Questions:**

1.	How is the optimal predefined number of subsequences/memories determined?

2.	How is the best top-k parameter selected?

3.	The intuition of using different memories to store different tokens is compelling. However, the pre-routing step before the input hidden representation is somewhat unclear. How can the router determine which memory to use without having access to the information stored in each memory, given that routing occurs before the key-value projection and memory update?

---

> ### Author Response · Authors · 2025-11-17
> **Response to Reviewer 51kq**
>
> We are very grateful for your positive and constructive review. We are encouraged that you appreciated our clear motivation and strong experimental evidence. Your questions about hyperparameter selection and the router's mechanics are very helpful for clarification.
>
> > `Weakness 1: Experiments with larger model sizes could further strengthen the paper.`
>
> Thank you for your valuable suggestion. Scaling the MoM architecture to larger models, and even exploring a MoM-MoE, is a very important and valuable direction for future work. In the current paper, we focused on introducing and validating the effectiveness of MoM at the **380M and 1.3B** scales. We are pleased to see that MoM's significant performance advantage over the baseline models is **consistent across both scales**, which demonstrates that the advantages of the MoM architecture have good scalability. We are confident in its potential in larger-scale models and will make this a priority for our next steps in research.
>
> > `Weakness 2, Q1 & Q2: Hyperparameter Choices (Memories and Top-k)`. Some hyperparameter choices—such as the top-k value in the routing process and the predefined number of memory slots—are not thoroughly discussed.
>
> Thank you for these important questions. These are indeed key hyperparameters. We did conduct detailed ablation studies on both, and we appreciate the opportunity to highlight those findings here, as they may not have been prominent enough in the original text.
>
> - **Regarding the number of memories (Q1)**: We explored this in our scaling experiment, presented in Figure 7. The results from that study indicate that performance on both Recall and Commonsense tasks improves as the number of memories increases from 1 to 8.
>
> - **Regarding the Top-k selection (Q2)**: This was analyzed in Figure 9 (Appendix H), which compares different activation ratios (k/n). The data from that experiment suggests that a 50% activation ratio (k/n=0.5) generally outperforms a 25% ratio.
>
> Therefore, based on this experimental data in the paper, we chose the **N=4, k=2** configuration as it provides a well-balanced trade-off between performance and efficiency.
>
> > `Question 3: Router Logic Before Memory Update`. How can the router determine which memory to use without having access to the information stored in each memory, given that routing occurs before the key-value projection and memory update?
>
> That is a very insightful question that gets to the heart of a fundamental design trade-off in modern sequence modeling. We thank the reviewer for this opportunity to clarify our rationale.
>
> Your question touches upon a classic intuition from traditional RNNs. Using the analogy of a forget gate, one might ask: "How can the model decide what to ***forget*** from memory without first ***knowing*** what is in the memory?" In a traditional, sequential RNN, the gate indeed depends on the previous state ($\textbf h_{t-1}$). This fundamental sequential dependency is the primary reason why traditional RNNs are difficult to parallelize over the time dimension. In contrast, many modern linear RNNs [1, 2], deliberately adopt state-independent gating that **relies only on the current input**, $\textbf x_t$. This design is crucial for enabling efficient, chunk-wise parallel training. Our router design adheres to this modern, parallel-friendly principle.
>
> While an ideal router might dynamically inspect the contents of each memory before making a decision, such a state-dependent approach would reintroduce the sequential bottleneck we aim to avoid. Therefore, our router **learns its policy from the input representation end-to-end**. It is not making a blind assignment. The final task loss propagates gradients back to the router's weights. This gradient signal effectively teaches the router whether assigning a given token $\textbf x_t$ to Memory 1 versus Memory 2 is more beneficial for minimizing the overall loss. Through this process, the router learns a specialized policy based entirely on the features of the input token.
>
> To empirically demonstrate the effectiveness of this learned strategy, we are happy to add a visualization analysis in the revised paper. As shown in the Figure 7, the routing decisions for tokens, projected into UMAP [3] space, **exhibit a clear clustering phenomenon**. This provides strong evidence that the router is not assigning tokens randomly. Instead, it has successfully learned to route tokens with similar features to the same memory. This behavior effectively creates a form of dynamic clustering at test time and confirms that **an input-dependent router can learn a highly efficient and meaningful routing policy**.
>
> We are truly grateful for your rigorous feedback! We hope our clarifications have better demonstrated our work.
>
> ---
>
> References:
>
> [1] Hierarchically gated recurrent neural network for sequence modeling
>
> [2] xlstm: Extended long short term memory
>
> [3] UMAP: Uniform Manifold Approximation and Projection for Dimension Reduction

---

> ### Author Response · Authors · 2025-11-24
> **Thank you & Looking Forward to Further Discussion**
>
> Dear Reviewer 51kq,
>
> We extend our deepest appreciation for the thorough and thoughtful review you provided for our manuscript. Your meticulous attention to detail is truly inspiring. To facilitate better communication and express our gratitude, we have carefully summarized our rebuttal below:
>
> - **Larger Model Scales.** `Weakness 1` We confirmed the robustness of MoM's gains across multiple scales (380M and 1.3B) and committed to larger-scale exploration in future work.
>
> - **Hyperparameter Choices** `Weakness 2 & Question 1,2` We clarified that our selected configuration ($N=4, k=2$) represents a favorable trade-off determined through our ablation studies (Figures 7 and 9), effectively balancing performance gains with efficiency.
>
> - **Router Mechanism & Dependency** `Question 3` We explained that our router uses state-independent and input-only gating to maintain parallel training efficiency, a standard practice in modern linear RNNs. We further added UMAP visualizations (Figure 5) to demonstrate that this input-dependent router successfully learns meaningful token clustering policies through end-to-end training.
>
> For further details regarding other issues, please consult our detailed rebuttal. We sincerely hope that our updates have addressed your concerns. We genuinely treasure your insightful comments as they have been pivotal in refining our paper. Thank you once again for your dedication, and we look forward to your feedback.
>
> Best regards,
>
> The Authors

---

### Official Review · Reviewer_SvM8 · 2025-10-30

**Soundness:** 3
**Presentation:** 3
**Contribution:** 2
**Rating:** 4
**Confidence:** 3

**Summary:**

This paper introduces Mixture-of-Memories (MoM), an architecture designed to enhance linear sequence models by addressing their fundamental limitation: compressing entire sequences into a single fixed-size memory state. MoM employs multiple independent memory states with a router network that directs input tokens to specific memories using a top-k selection mechanism. The approach is inspired by theta-gamma oscillations in the hippocampus and borrows conceptually from Mixture-of-Experts (MoE) architectures. The authors demonstrate that MoM maintains O(n) training complexity and O(1) inference complexity while significantly improving performance on recall-intensive tasks. Experiments on 380M and 1.3B parameter models show MoM outperforms existing linear sequence models and approaches Transformer performance on recall tasks, though with modest increases in parameter count. The work provides a general framework compatible with diverse memory update mechanisms and includes a hardware-efficient implementation using varlen operations.

**Strengths:**

1. Well-motivated problem formulation: The paper clearly identifies memory interference and limited capacity as core issues in linear sequence models, providing both intuitive explanations and neuroscience-inspired motivation from hippocampal theta-gamma oscillations.
2. General and flexible framework: MoM's compatibility with various memory update mechanisms (Table 1 lists 11 different methods) demonstrates its generality. This is a significant practical advantage, allowing easy integration with future linear modeling innovations.
3. Comprehensive experimental evaluation: The paper includes extensive experiments across multiple benchmarks (recall-intensive tasks, long-context tasks, commonsense reasoning), multiple model scales (380M, 1.3B parameters), and thorough ablations (memory count, activation ratios, auxiliary loss, shared memory).
4. Strong empirical results on target tasks: MoM achieves substantial improvements on recall-intensive tasks (28.16 vs 24.78 average for Gated DeltaNet at 380M; 36.04 vs 32.30 at 1.3B), effectively narrowing the gap with Transformers (31.70 and 37.31 respectively).
5. Hardware-efficient implementation: Section 3.3 provides a mathematically rigorous description of the varlen-based implementation, demonstrating practical viability. Figure 3 shows clear efficiency advantages over Transformers at long sequences.

**Weaknesses:**

1. While related work in linear sequence modeling and MoE has been mentioned, it lacks in-depth comparisons and differential analyses with recent methods (such as RWKV-7 and Titans). In particular, Table 1 lists these methods but does not systematically experimentally compare the performance differences of MoM under different memory update mechanisms.
2. Table 5 claims to have discovered "specialization" in different memories, but it is based solely on qualitative observations of an intermediate layer and lacks systematic and statistical significance verification. This "discovery" is more like cherry-picking data than rigorous analysis.
3. The authors claim that the core advantage of MoM is "eliminates memory interference" by separating memory states, but also introducing shared memory to capture global information. Does this contradict the core design? In particular, in the ablation of table 6, there seems to be a lack of settings for "shared memory only" and "neither shared memory nor single memory exists". This lack of ablation experiment is fatal because it directly relates to whether "mixture" is truly necessary.

**Questions:**

1. Could shared memory become a new bottleneck? If all tokens access shared memory, will its capacity quickly become saturated? Have you analyzed the actual utilization of shared memory across different layers?
2. More details about reproduction：
	a. What is the computational overhead of the router network?
	b. How sensitive is performance to router initialization?
	c. What are actual wall-clock training times compared to baselines?

---

> ### Author Response · Authors · 2025-11-17
> **[Part 1] Response to Reviewer SvM8**
>
> We sincerely thank you for your exceptionally thorough and rigorous review. We are truly grateful for your detailed acknowledgment of our strengths from motivation to implementation and for your critical and precise feedback on our ablation studies and analyses. Your insights are invaluable for strengthening this work. We provide our responses below.
>
> > `Weakness 1: On Comparisons with Recent Methods and Across Memory Mechanisms.` ...lacks in-depth comparisons... with recent methods (such as RWKV-7 and Titans)... and does not systematically experimentally compare the performance differences of MoM under different memory update mechanisms.
>
> Thank you for this point. We frame our work within the mainstream view of linear models as Test-Time Training (TTT) systems [1,2,3]. In this view, the memory $\mathbf{M}$ is dynamically fit to a $\mathbf{k} \to \mathbf{v}$ mapping (optimizing $\mathbf{k}\mathbf{M}=\mathbf{v}$), primarily following two major learning strategies:
>
> 1. **The Hebbian Rule** [4]: This strategy reinforces the $\mathbf{k}-\mathbf{v}$ association, optimizing an objective like $\langle \mathbf{k}_t\mathbf{M}_t, \mathbf{v}_t \rangle$. We selected GLA as its representative, which adds a regularization term to the objective: $\langle \mathbf{k}_t\mathbf{M}_t, \mathbf{v}_t \rangle + \frac{\alpha_t}{2} \|\mathbf{M}_t\|_2^2$ to avoid overfitting.
> 2. **The Delta Rule** [5]: This strategy is based on error correction, using an L2 loss $\|\mathbf{v}_t - \mathbf{k}_t\mathbf{M}_t\|_2^2$. We selected Gated Deltanet as its representative. It also introduces regularization, $\beta_t \|\mathbf{v}_t - \mathbf{k}_t\mathbf{M}_t\|_2^2 + \frac{\alpha_t}{2} \|\mathbf{M}_t\|_2^2$, and its memory update mechanism is $\mathbf{M}\_t = \alpha\_t(\mathbf{I} - \mathbf{k}^T\_t\mathbf{k}\_t)\mathbf{M}\_{t-1} + \beta\_t\mathbf{k}^T\_t\mathbf{v}\_t$.
>
> RWKV-7 and Titans, are both excellent variants based on the Delta Rule. RWKV-7's main difference is replacing scalar gates with diagonal matrices for finer-grained gating, while Titans improves the Delta Rule's optimizer by introducing momentum.
>
> Therefore, by conducting experiments on GLA and Gated Deltanet, two models representing these fundamentally different learning strategies, we have validated the effectiveness of the MoM architecture. As shown in Table 4, MoM delivers consistent average improvements on both. This demonstrates the generality and effectiveness of the MoM framework itself, confirming its benefits are not limited to a single update mechanism.
>
> | Model | Params | FDA | SWDE | SQUAD | NQ | TriviaQA | Drop | Avg. |
> | :--- | :--- | :--- | :--- | :--- | :--- | :--- | :--- | :--- |
> | GLA *expanded* | 425M | **15.08** | 20.15 | 28.28 | 13.30 | 41.65 | 18.74 | 22.87 |
> | GLA *MoM* | 395M | 9.90 | **21.65** | **29.36** | **14.16** | **45.20** | **20.89** | **23.53** |
> ||
> | Gated DeltaNet *expanded* | 550M | 18.26 | 24.27 | **30.03** | **17.74** | 48.34 | 19.26 | 26.32 |
> | Gated DeltaNet *MoM* | 444M | **22.98** | **29.90** | 29.69 | 16.60 | **48.82** | **20.99** | **28.16** |
>
> ---
>
> References:
>
> [1] Learning to (Learn at Test Time): RNNs with Expressive Hidden States
>
> [2] Speed Always Wins: A Survey on Efficient Architectures for Large Language Models
>
> [3] Test-time regression: a unifying framework for designing sequence models with associative memory
>
> [4] Concepts of Soft Computing: Fuzzy and ANN with Programming, pages 175–182, 2019.
>
> [5] Neural network capacity using delta rule

---

> > ### Comment · Reviewer_SvM8 · 2025-11-25
> >
> > It seems that for this question, no additional experiments were added. Instead, a more detailed classification of the baseline was made, and the provided result was to illustrate the effectiveness of each  baseline. For the remaining parts, I am overall sastified with the provided responses. Please add them in the revised version. Considering other reviews all together, I would change my score.

---

> ### Author Response · Authors · 2025-11-17
> **[Part 2] Response to Reviewer SvM8**
>
> > `Weakness 2: Qualitative and Unrigorous Specialization Analysis.` Table 5 claims... "specialization"... but it is based solely on qualitative observations... This "discovery" is more like cherry-picking data than rigorous analysis.
>
> We sincerely thank you for your insightful critique regarding the memory specialization analysis in Table 5. We agree that the original GPT-based part-of-speech analysis was indeed qualitative and rather vague. In response to your concern, we have **removed this section in the Revision**. In its place, we have added a **new, more rigorous quantitative analysis** to objectively explore the specialization trends in MoM.
>
> Specifically, we collected the hidden states of all tokens routed to each memory while the MoM model was performing inference on the ARC-Easy task. We then used **UMAP** [6] to project these high-dimensional vectors and visualized them, coloring each token by its Top-1 target memory. We kindly refer the reviewer to **Figure 5 of the Revision** for this clustering effect. We are pleased to report that in the deeper layers of the model, the token vectors routed to different memories form **distinct clusters** in the UMAP space. This confirms that the router is performing a non-random sorting, having learned to route tokens with similar features to the same memory. We note that some overlap exists between clusters. This is expected, as our model uses k=2 routing (sending information to two memories), but our visualization only shows the Top-1 memory for clarity. Thus, overlapping boundaries are consistent with our design.
>
> This clustering analysis provides a new perspective for understanding MoM through the lens of **Test-Time Training** (TTT) [1,2,3]. Our memory update mechanism (Gated Deltanet) employs a Delta Rule learning style, dynamically fitting a $\mathbf{k} \to \mathbf{v}$ mapping at test time (optimizing $\mathbf{k}\mathbf{M}=\mathbf{v}$). When test data is feature-sparse and widely distributed, a single memory network $\mathbf{M}$ struggles to fit all data points quickly and accurately. The UMAP analysis demonstrates that the MoM router acts as a dynamic clustering mechanism. It automatically partitions the broad input stream into multiple, more concentrated, and feature-cohesive subsets at inference time, handing each subset to a specialized memory (e.g., $\mathbf{M}_2$). This realizes a mechanism analogous to **ensemble learning within the TTT framework**: each memory $\mathbf{M}_m$ no longer needs to fit the entire, complex data distribution, but only a simpler sub-distribution. This reduces the learning difficulty for each memory.
>
> We thank you again for this valuable question. It motivated us to dig deeper into the working mechanisms of MoM.
>
> ---
>
> References:
>
> [1] Learning to (Learn at Test Time): RNNs with Expressive Hidden States
>
> [2] Speed Always Wins: A Survey on Efficient Architectures for Large Language Models
>
> [3] Test-time regression: a unifying framework for designing sequence models with associative memory
>
> [6] UMAP: Uniform Manifold Approximation and Projection for Dimension Reduction

---

> ### Author Response · Authors · 2025-11-17
> **[Part 3] Response to Reviewer SvM8**
>
> > `Weakness 3: On the Role of Shared Memory and Ablation Completeness`. Does this introducing shared memory contradict the core design? ...in the ablation of table 6, there seems to be a lack of settings for "shared memory only"... This lack of ablation experiment is fatal because it directly relates to whether "mixture" is truly necessary.
>
> Thank you for raising this critical point, which we agree is fundamental to understanding the contribution of our work. We are happy to clarify the design and provide the specific ablation data you requested.
>
> We agree that the Shared Memory is indeed important. As we discussed in Section 3.2.2, the split memories only observe partial token sequences and store local information. Therefore, a shared memory that is routed to by all tokens is necessary to capture complete, global information. This does not conflict with our mixture mechanism; rather, it is a **complementary design**, similar to "shared experts" widely used in modern MoE works [7,8].
>
> Regarding the performance gains, please note that if we only keep the Shared Memory, our MoM model degrades to the standard Gated Deltanet baseline. The paper already contains the key ablation experiment you pointed to, and **we sincerely apologize for not making this comparison explicit**. We present the direct comparison in the table below. As you can see, the "Split Memory Only" model still outperforms the "Shared Memory Only" baseline.
>
> | Model Configuration | Avg. (Recall) |
> | :--- | :---: |
> | Mixed Memory | **28.16** |
> | Split Memory Only | 26.06 |
> | Shared Memory Only | 24.78 |
>
> Furthermore, as shown in **Section 4.2.3** (Table 4), we also compared MoM to an **enlarged single memory which is also "shared"**. The results confirm that MoM still performs better, which validates that the mixture mechanism of routing to multiple independent memories is indeed necessary for the performance gains.
>
> | Model | Params | FDA | SWDE | SQUAD | NQ | TriviaQA | Drop | Avg. |
> | :--- | :--- | :--- | :--- | :--- | :--- | :--- | :--- | :--- |
> | GLA *expanded* | 425M | **15.08** | 20.15 | 28.28 | 13.30 | 41.65 | 18.74 | 22.87 |
> | GLA *MoM* | 395M | 9.90 | **21.65** | **29.36** | **14.16** | **45.20** | **20.89** | **23.53** |
> ||
> | Gated DeltaNet *expanded* | 550M | 18.26 | 24.27 | **30.03** | **17.74** | 48.34 | 19.26 | 26.32 |
> | Gated DeltaNet *MoM* | 444M | **22.98** | **29.90** | 29.69 | 16.60 | **48.82** | **20.99** | **28.16** |
>
> > `Question 1: Shared Memory as a Potential Bottleneck and its Utilization.` If all tokens access shared memory, will its capacity quickly become saturated? Have you analyzed the actual utilization of shared memory across different layers?
>
> Your intuition is absolutely correct. A single shared memory would be a bottleneck, and our experimental data confirms this.
>
> The strongest evidence is in **Table 4**. The Single-Enlarged model represents this exact scenario of a single shared memory forced to process all tokens. Its average performance is significantly weaker than our MoM, which achieves a higher average score despite having **fewer parameters**. This performance gap empirically validates that the single memory bottleneck.
>
> However, in our complete MoM architecture, the "Mixture" mechanism and the "Shared Memory" **work in synergy**. Our UMAP visualization analysis in the Revision also shows that the mixture memories offload the Shared Memory's burden by handling highly specialized, feature-distinct token clusters. This allows the Shared Memory to focus on capturing the global context, which it excels at, without becoming saturated by all token types.
>
> Our experimental data supports this synergistic view. Both the Mixture-only model and the Shared Memory-only baseline are suboptimal. Only their combination achieves the best performance. Therefore, the Shared Memory does not become a bottleneck in MoM, because one of the Mixture mechanism's functions is precisely to prevent it from becoming one.
>
> We have further evidence to support this conclusion. As shown in **Figures 7 and 9**, while the Shared Memory is always present, the model's performance **scales positively with the number of split memories**. If the Shared Memory were the bottleneck, performance would quickly plateau. The fact that it does not proves the bottleneck is not the Shared Memory.
>
> As for analyzing its utilization across different layers, that is an excellent and valuable suggestion! We will make this a priority for our future work.
>
> ---
>
> References:
>
> [7] Qwen3 Technical Report
>
> [8] The Llama 4 herd: The beginning of a new era of natively multimodal AI innovation

---

> ### Author Response · Authors · 2025-11-17
> **[Part 4] Response to Reviewer SvM8**
>
> > `Question 2 (a, c): Computational Overhead`. What is the computational overhead of the router network? What are actual wall-clock training times compared to baselines?
>
> We conducted detailed throughput tests and performance profiling. By measuring the time of each computation in the MoM layer, we observed that the **router's overhead is very small**, accounting for **less than 2%** of the total MoM layer time. We tested the actual training throughput and found the Gated DeltaNet baseline runs at 108,134 tokens/sec, while MoM runs at 42,599 tokens/sec. The actual wall-clock training times correspond to this.
>
> This overhead is introduced because our varlen implementation needs to perform complex token reordering and reorganization to efficiently process multiple memory streams in parallel on a GPU. Similar operations in related work, such as MoBA [9], also lead to a decrease in efficiency. Our specific profiling shows the total MoM layer overhead is around 16ms, of which the GDN operator itself takes ~3.2ms, while the **token reordering process** takes ~5.98ms. This PyTorch-based reordering is the main performance bottleneck. This efficiency drop is a common issue in routing-based mechanisms, and drawing inspiration from MoE optimizations, it can be mitigated through techniques like expert parallelism. As MoM is an exploratory work, we will focus on further optimizing this overhead for large-scale industrial use in the future.
>
> > `Question 2 (b): How sensitive is performance to router initialization?`
>
> We are grateful for this thoughtful question. To test the effect of different router initializations, we conducted an ablation study. We fixed the initialization seed for all other model weights to 42 and only used **different initialization seeds for the router**. Due to limited computational resources, we ran this test on the 380M model, training for 5B tokens, to verify the router's robustness.
>
> The experimental results show that different initializations had a negligible impact on the final task loss, with the variance remaining within 0.07%. We also tested the performance on commonsense tasks and found **no significant differences**, confirming that our model is robust to router initialization.
>
> | Router Seed | loss | avg. |
> | --- | --- | --- |
> | 1 | 2.9396 | 40.37 |
> | 42 | 2.9409 | 40.20 |
> | 1234 | 2.9426 | 40.07 |
>
> We sincerely thank you for your thorough and critical review. We hope our clarifications and additional data have fully resolved your concerns. If you now find our approach more convincing, we would be deeply grateful if you might possibly kindly consider revisiting the evaluation of our manuscript. Thank you immensely!
>
> ---
>
> References:
>
> [9] MoBA: Mixture of Block Attention for Long-Context LLMs

---

> ### Author Response · Authors · 2025-11-24
> **Thank you & Looking Forward to Further Discussion**
>
> Dear Reviewer SvM8,
>
> We extend our deepest appreciation for the thorough and thoughtful review you provided for our manuscript. Your meticulous attention to every detail is truly inspiring. To facilitate better communication and express our gratitude, we have carefully summarized our rebuttal below:
>
> - **Comparison with Recent Methods.** `Weakness 1` We clarified that MoM is a general framework validated on both Hebbian-like and Delta Rule based models, which covers the fundamental mechanisms behind recent works like RWKV-7 and Titans.
>
> - **Quantitative Memory Specialization.** `Weakness 2` Inspired by your feedback, we conducted a rigorous quantitative UMAP analysis (Figure 5 in Revision). The results reveal clear clustering of tokens routed to specific memories in deeper layers, demonstrating memory specialization. We interpret this as a form of ensemble learning within the Test-Time Training (TTT) framework.
>
> - **Shared Memory & Ablation.** `Weakness 3 & Qustion 1` We highlighted the comparison in Table 4 between MoM and the single expanded memory, which validates the effectiveness of the split-and-mixture mechanism. The scaling results in Figure 7 further show that the shared component is not the sole source of gains nor a bottleneck in our design.
>
> - **Computational Overhead & Robustness.** `Qustion 2` We reported detailed profiling results, showing the router overhead is minimal (<2%) and confirmed the router's insensitivity to initialization.
>
> For other issues not mentioned here, please refer to our detailed rebuttal response. We sincerely hope this addresses your concerns! We humbly look forward to further discussion with you. Your understanding means the world to us.
>
> Warm regards,
>
> Authors

---

> ### Author Response · Authors · 2025-11-25
> **Deeply Grateful for Your Recognition and the Score Update!**
>
> Dear Reviewer SvM8,
>
> We are truly thrilled to receive your positive feedback and your decision to raise the score! It gives us tremendous encouragement to know that our responses have satisfactorily addressed your concerns. We have updated the revised manuscript following your valuable guidance.
>
> Thank you once again for your rigorous review and for guiding us to make this paper better. Your support means the world to us!
>
> Warm regards,
>
> Authors

---

### Official Review · Reviewer_EVGR · 2025-10-30

**Soundness:** 3
**Presentation:** 2
**Contribution:** 3
**Rating:** 6
**Confidence:** 4

**Summary:**

This paper addresses a well-known weakness of linear sequence models (such as linear RNNs and state-space models): their reliance on a single, fixed-size memory state, which leads to memory interference and poor performance on recall-intensive tasks. The authors propose Mixture-of-Memories (MoM), a new architecture that utilizes multiple independent memory states. A top-k routing network directs each input token to a specific subset of these memories for updating. This mechanism aims to increase the model's effective memory capacity and reduce interference by allowing different types of information to be stored in separate states. The final output is computed by querying a weighted mixture of the activated memory states. The paper demonstrates through extensive experiments that MoM significantly outperforms strong linear model baselines on a variety of downstream tasks, particularly those requiring information recall. Notably, the model achieves performance comparable to standard Transformer models while retaining the linear-time training and constant-time inference benefits of linear RNNs.

**Strengths:**

- The paper targets a critical and widely recognized weakness of linear-time sequence models: their poor performance on recall-intensive tasks due to the bottleneck of a single fixed-size memory state.
- The experimental results are comprehensive and robust.

- The core idea of using a top-k router to manage multiple, independent RNN memory states is a novel and clever application of sparse activation principles.

**Weaknesses:**

- The ablation study in Table 6 indicates that the "Shared Memory" component is critical. Removing it causes a large performance drop (e.g., 2.1 points on Recall tasks). This makes it difficult to disentangle the gains from the "Mixture" mechanism (routing to $k$ of $N$ memories) from the gains of simply having a parallel, always-on "Shared Memory" state. The paper's narrative focuses heavily on the top-k mixture, but a large portion of the gains might be coming from this simpler shared component.
- The main results in Table 2 appear to compare models of slightly different sizes. For example, Table 4 shows the "Gated DeltaNet MoM" model has 444M parameters, while its baseline "Gated DeltaNet" is 380M. This ~17% parameter difference complicates the main comparison. While a fairer study with equal activated parameters is commendably included in Appendix G, the primary results table in the main body of the paper should ideally be based on a more strictly controlled comparison.
- The paper repeatedly claims MoM is a "new paradigm" and "fundamentally differs" from MoE (e.g., in Appendix B). This distinction feels overstated. The core mechanism, a top-k router that sparsely activates a subset of modules, which is the defining feature of MoE. Applying this concept to RNN states and key/value projections instead of FFN layers is a novel and useful application, but it does not seem to be a fundamental departure from the MoE paradigm.
-  The analysis of memory specialization in Section 4.2.6 and Table 5 is weak. The paper provides no quantitative methodology for how token categories (e.g., "Basic nouns/verbs" vs. "Proper nouns/scientific terms") were defined or how the "Potential Function" (e.g., "Simplify semantic information") was determined. As presented, these conclusions appear entirely subjective and anecdotal, lacking the empirical rigor demonstrated in the rest of the paper.

**Questions:**

See my weakness part.

---

> ### Author Response · Authors · 2025-11-17
> **[Part 1] Response to Reviewer EVGR**
>
> We are grateful for your insightful comments and valuable suggestions. Your feedback has been highly instructive and has helped us to significantly strengthen our paper. We provide our responses below.
>
> > `Weakness 1: Disentangling Gains from Shared Memory vs. Mixture Mechanism.` The ablation study in Table 6 indicates that the "Shared Memory" component is critical. This makes it difficult to disentangle the gains from the "Mixture" mechanism from the gains of simply having a parallel, always-on "Shared Memory" state.
>
> Thank you for this sharp observation. We agree that the Shared Memory component is indeed critical, as shown in Table 6. As we discussed in Section 3.2.2, this is by design. The separated memories only observe a subset of tokens and thus store local information. Therefore, a shared component that is routed to by all tokens is necessary to capture complete, global information.
> This design is complementary to our mixture mechanism, not in conflict with it. In fact, similar "shared experts" are a common and effective component in many modern MoE architectures [1, 2].
>
> We sincerely apologize for not making this comparison more explicit in our original submission. Allow us to disentangle the source of the gains here:
>
> 1. Please note that if we only keep the Shared Memory and remove the mixture part, the model simply **degrades to the standard Gated Deltanet baseline**.
>
> 2. The ablation in Table 6 also shows that the "w/o Shared Memory" model, which only has the separated memories, **still outperforms the standard Gated Deltanet baseline**. This confirms the mixture mechanism itself provides significant value.
>
> | Model Configuration | Avg. (Recall) |
> | :--- | :---: |
> | Mixed Memory | **28.16** |
> | Split Memory Only | 26.06 |
> | Shared Memory Only | 24.78 |
>
> Finally, and most importantly, our control experiment presented in **Section 4.2.3** and **Table 4** directly addresses this. We compared MoM to a single expanded memory baseline, which is by definition a shared memory. The results clearly show that our full MoM model outperforms this **enlarged single shared memory**. This provides the evidence that the performance gain originates from the mixture and routing mechanism itself, and not just from the presence of a shared component.
>
> | Model | Params | FDA | SWDE | SQUAD | NQ | TriviaQA | Drop | Avg. |
> | :--- | :--- | :--- | :--- | :--- | :--- | :--- | :--- | :--- |
> | GLA *expanded* | 425M | **15.08** | 20.15 | 28.28 | 13.30 | 41.65 | 18.74 | 22.87 |
> | GLA *MoM* | 395M | 9.90 | **21.65** | **29.36** | **14.16** | **45.20** | **20.89** | **23.53** |
> ||
> | Gated DeltaNet *expanded* | 550M | 18.26 | 24.27 | **30.03** | **17.74** | 48.34 | 19.26 | 26.32 |
> | Gated DeltaNet *MoM* | 444M | **22.98** | **29.90** | 29.69 | 16.60 | **48.82** | **20.99** | **28.16** |
>
> > `Weakness 2: Fairness of Parameter Comparison in Main Results.` The main results in Table 2 appear to compare models of slightly different sizes... while a fairer study is commendably included in Appendix G.
>
> We thank the reviewer for raising this important point about parameter fairness. As the reviewer commendably noted, we did provide a strictly controlled comparison in Appendix G (Table 9). In that fair comparison, MoM still achieves a higher average score across all tasks than the baseline. This confirms that the advantages of MoM do not stem from a simple increase in parameters.
>
> To eliminate this ambiguity, we have **moved Table 9 from the appendix into the main body** in the Revision to serve as our primary results table.
>
> > `Weakness 3: Overstated Distinction from the MoE Paradigm.`
>
> Thank you for your valuable feedback on this. Our intent was to emphasize the different implications of applying this mechanism to RNN states versus FFN layers. An MoE on an FFN layer is often context-independent, serving as a specialized linear mapping. In contrast, when applied to RNN states, each memory is updated by a context-dependent subset of tokens.
>
> Furthermore, these multiple independent memories can interact in various ways. The weighted sum (which is analogous to MoE) is just one such method, and **many other interesting interactions are worth exploring**. In the Revision, we have **modified the relevant wording and removed the inappropriate claims** to more accurately position our work.
>
> ---
>
> References:
>
> [1] Qwen3 Technical Report
>
> [2] The Llama 4 herd: The beginning of a new era of natively multimodal AI innovation

---

> ### Author Response · Authors · 2025-11-17
> **[Part 2] Response to Reviewer EVGR**
>
> > `Weakness 4: Memory Specialization Analysis and Quantitative Measures.` The analysis... in Section 4.2.6 and Table 5 is weak... The paper provides no quantitative methodology... these conclusions appear entirely subjective and anecdotal.
>
> We sincerely thank you for your insightful critique regarding the memory specialization analysis in Table 5. We agree that the original GPT-based part-of-speech analysis was indeed qualitative and rather vague. In response to your concern, we have **removed this section in the Revision**. In its place, we have added a **new, more rigorous quantitative analysis** to objectively explore the specialization trends in MoM.
>
> Specifically, we collected the hidden states of all tokens routed to each memory while the MoM model was performing inference on the ARC-Easy task. We then used **UMAP** [3] to project these high-dimensional vectors and visualized them, coloring each token by its Top-1 target memory. We kindly refer the reviewer to **Figure 5 of the Revision** for this clustering effect. We are pleased to report that in the deeper layers of the model, the token vectors routed to different memories form **distinct clusters** in the UMAP space. This confirms that the router is performing a non-random sorting, having learned to route tokens with similar features to the same memory. We note that some overlap exists between clusters. This is expected, as our model uses k=2 routing (sending information to two memories), but our visualization only shows the Top-1 memory for clarity. Thus, overlapping boundaries are consistent with our design.
>
> This clustering analysis provides a new perspective for understanding MoM through the lens of **Test-Time Training** (TTT) [4,5]. Our memory update mechanism (Gated Deltanet) employs a Delta Rule [6] learning style, dynamically fitting a $\mathbf{k} \to \mathbf{v}$ mapping at test time (optimizing $\mathbf{k}\mathbf{M}=\mathbf{v}$). When test data is feature-sparse and widely distributed, a single memory network $\mathbf{M}$ struggles to fit all data points quickly and accurately. The UMAP analysis demonstrates that the MoM router acts as a dynamic clustering mechanism. It automatically partitions the broad input stream into multiple, more concentrated, and feature-cohesive subsets at inference time, handing each subset to a specialized memory (e.g., $\mathbf{M}_2$). This realizes a mechanism analogous to **ensemble learning within the TTT framework**: each memory $\mathbf{M}_m$ no longer needs to fit the entire, complex data distribution, but only a simpler sub-distribution. This reduces the learning difficulty for each memory.
>
> We would like to sincerely thank you again for your inspiring discussion. We humbly hope that our responses have addressed your points adequately.
>
> ---
>
> References:
>
> [3] UMAP: Uniform Manifold Approximation and Projection for Dimension Reduction
>
> [4] Learning to (Learn at Test Time): RNNs with Expressive Hidden States
>
> [5] Test-time regression: a unifying framework for designing sequence models with associative memory
>
> [6] Neural Network Capacity Using Delta Rule

---

> ### Author Response · Authors · 2025-11-24
> **Thank you & Looking Forward to Further Discussion**
>
> Dear Reviewer EVGR,
>
> We sincerely thank you for your insightful review. Your feedback was instrumental in helping us identify and fix key presentation issues. For clarity, we have summarized our main responses and revisions below:
>
> - **Gains Disentanglement.** `Weakness 1` We clarified through ablation and control experiments that the mixture mechanism itself provides significant gains beyond just the shared component.
>
> - **Quantitative Memory Specialization.** `Weakness 4` Inspired by your feedback, we conducted a rigorous quantitative UMAP analysis (Figure 5 in Revision). The results reveal clear clustering of tokens routed to specific memories in deeper layers, demonstrating memory specialization. We interpret this as a form of ensemble learning within the Test-Time Training (TTT) framework.
>
> - **Paper Presentation.** `Weakness 2,3` We moved the fair comparison table to the main text and refined MoM's positioning to align with the MoE paradigm, ensuring a rigorous and accurate narrative.
>
> For other details not mentioned in this summary, please kindly refer to our full rebuttal response. The depth of your review has been a great source of inspiration for us and we deeply appreciate your effort. We look forward to the opportunity for further engagement with you.
>
> Best regards,
>
> Authors

---

### Official Review · Reviewer_vGVV · 2025-11-01

**Soundness:** 3
**Presentation:** 3
**Contribution:** 3
**Rating:** 6
**Confidence:** 2

**Summary:**

This paper presents Mixture-of-Memories (MoM), a framework for linear sequence modeling that replaces a single fixed memory with multiple independent ones combined through a routing network. Drawing inspiration from both neuroscience (multi-item memory mechanisms) and the mixture-of-experts (MoE) concept, MoM expands memory capacity and reduces interference, making it especially effective for recall-intensive tasks. Experiments on language benchmarks show that MoM delivers clear gains in recall-focused and long-context tasks, often matching or surpassing Transformer baselines in performance.

**Strengths:**

1. The paper pinpoints the weakness of compressing an entire sequence into a single memory state in linear models and connects this limitation to memory interference.

2. The empirical study is extensive, covering recall-intensive benchmarks, long-context tasks, memory behavior analysis, scaling and ablation studies.

3. The authors provide qualitative insights into memory specialization and routing distributions, enhancing the interpretability of MoM.

4. Implementation details show careful consideration of computational cost, employing Triton kernels for batched memory routing and maintaining O(n) training and O(1) inference efficiency.

**Weaknesses:**

1. The router formulation specifies a scheme combining softmax and Top-K but lacks crucial implementation details. It does not clarify how ties in Top-K selection are resolved, whether gradients are propagated through the Top-K operation or treated as non-differentiable, or how the learned matrix affects the sparsity of routing and the memory load balance. It also leaves open whether the router is robust to input distribution shifts or token imbalance.

2. Table 5 provides qualitative insights into memory specialization, but the analysis remains descriptive. Quantitative measures would better substantiate claims of interpretability.

3. The theta-gamma memory analogy provides an appealing motivation but lacks a rigorous computational mapping. The biological discussion lack ablation or formal modeling to validate the link between oscillatory cycles and MoM’s memory dynamics.

**Questions:**

1. Could the authors clarify how gradients are propagated through the Top-K router?

2. For recall-intensive benchmarks, the source of improvement is unclear. It is not specified whether the gains come from increased memory capacity, larger parameter count, or better routing dynamics. Could a control experiment with a single enlarged memory help isolate these factors?

3. In the hybrid architecture experiments, do the observed gains mainly result from increased depth, or is there measurable synergy between global attention and mixture-based memory modules?

---

> ### Author Response · Authors · 2025-11-17
> **[Part 1] Response to Reviewer vGVV**
>
> We sincerely thank you for your thoughtful review and for recognizing the strengths of our work, including the extensive empirical study and interpretability analysis. Your insightful questions regarding the router implementation and the source of performance gains are highly valuable. We address these points below.
>
> > `Weakness 1 & Question 1: Router Implementation Details and Gradient Propagation.` The router formulation... lacks crucial implementation details. Could the authors clarify how gradients are propagated through the Top-K router?
>
> We thank the reviewer for these crucial questions. We clarify the router's training process by addressing two key aspects: (1) the gradient from the main task loss, and (2) the auxiliary loss for load balancing.
>
> - **Task Gradient Propagation**: As detailed in Section 3.2.1, the router's scores, $g_t$, are used as weighting coefficients to mix the memory states: $\tilde{\textbf M}_t = \sum_m g_t^{(m)} \textbf M_t^m$. This mixed state $\tilde{\textbf M}_t$ directly participates in the final output $\textbf o_t$ calculation. Consequently, the gradient from the final task loss propagates back to $g_t$ through $\textbf o_t$ and $\tilde{\textbf M}_t$.Critically, since $g_t$ is determined by the TopK selection, the gradient only flows through the $K$ paths that were selected. The gradients for the $(M-K)$ unselected memories are zero for that specific step. We also note our softmax-then-TopK design (rather than TopK-then-softmax), which ensures that gradients from all activated paths are back-propagated to the entire routing weight matrix. This process effectively trains the router to select the optimal combination of memories to minimize the task loss.
> - **Load Balancing and Robustness**: Beyond the task loss, we introduce an auxiliary load balancing loss (Section 4.2.6) to prevent the router from starving certain memories by consistently favoring others [1]. This auxiliary loss incentivizes the router to distribute tokens as evenly as possible across all memories. The effectiveness of this approach is validated in our ablation study (Table 6) and visualizations (Figures 6 and 10), which show the token distribution per memory during inference. Without the auxiliary loss, we observe clear memory degradation. With its inclusion, the memory load is significantly more balanced, leading to a more robust model.
>
> The overall training objective for the router is a weighted sum of these two components, where a hyperparameter controls the influence of the auxiliary loss. The importance of this balance is validated in our ablation study in Table 6.
>
> ---
>
> References:
>
> [1] Switch Transformers: Scaling to Trillion Parameter Models with Simple and Efficient Sparsity

---

> ### Author Response · Authors · 2025-11-17
> **[Part 2] Response to Reviewer vGVV**
>
> > `Weakness 2: Memory Specialization Analysis and Quantitative Measures.` Table 5 provides qualitative insights into memory specialization, but the analysis remains descriptive. Quantitative measures would better substantiate claims of interpretability.
>
> We sincerely thank you for your insightful critique regarding the memory specialization analysis in Table 5. We agree that the original GPT-based part-of-speech analysis was indeed qualitative and rather vague. In response to your concern, we have **removed this section in the Revision**. In its place, we have added a **new, more rigorous quantitative analysis** to objectively explore the specialization trends in MoM.
>
> Specifically, we collected the hidden states of all tokens routed to each memory while the MoM model was performing inference on the ARC-Easy task. We then used **UMAP** to project these high-dimensional vectors and visualized them, coloring each token by its Top-1 target memory. We kindly refer the reviewer to **Figure 5 of the Revision** for this clustering effect. We are pleased to report that in the deeper layers of the model, the token vectors routed to different memories form **distinct clusters** in the UMAP space. This confirms that the router is performing a non-random sorting, having learned to route tokens with similar features to the same memory. We note that some overlap exists between clusters. This is expected, as our model uses k=2 routing (sending information to two memories), but our visualization only shows the Top-1 memory for clarity. Thus, overlapping boundaries are consistent with our design.
>
> This clustering analysis provides a new perspective for understanding MoM through the lens of **Test-Time Training** (TTT) [2]. Our memory update mechanism (Gated Deltanet) employs a Delta Rule [3] learning style, dynamically fitting a $\mathbf{k} \to \mathbf{v}$ mapping at test time (optimizing $\mathbf{k}\mathbf{M}=\mathbf{v}$). When test data is feature-sparse and widely distributed, a single memory network $\mathbf{M}$ struggles to fit all data points quickly and accurately. The UMAP analysis demonstrates that the MoM router acts as a dynamic clustering mechanism. It automatically partitions the broad input stream into multiple, more concentrated, and feature-cohesive subsets at inference time, handing each subset to a specialized memory (e.g., $\mathbf{M}_2$). This realizes a mechanism analogous to **ensemble learning within the TTT framework**: each memory $\mathbf{M}_m$ no longer needs to fit the entire, complex data distribution, but only a simpler sub-distribution. This reduces the learning difficulty for each memory.
>
> We thank you again for this valuable question. It motivated us to dig deeper into the working mechanisms of MoM.
>
> > `Weakness 3: Theta-Gamma Analogy.` The theta-gamma memory analogy provides an appealing motivation but lacks a rigorous computational mapping...
>
> We thank you for this feedback. The biological memory analogy was intended as a high-level inspiration, primarily to heuristically introduce the problem. We agree that it is not a rigorous computational mapping. We will tone down the wording in the Revision to make this distinction clear.
>
> > `Question 2: Source of Improvement and Control Experiment.` It is not specified whether the gains come from... Could a control experiment with a single enlarged memory help isolate these factors?
>
> Thank you for this excellent question, which touches on the very essence of MoM's contribution. This is precisely the control experiment we conducted in **Section 4.2.3** and presented in **Table 4**.
>
> To isolate these factors, we compared MoM against a baseline with a single memory enlarged to match the total capacity of our MoM model. We validated this on both GLA and Gated Deltanet, which represent two fundamentally different learning strategies: the **Hebbian Rule** and the **Delta Rule**, respectively. The results clearly show that MoM consistently outperforms the single expanded memory model, despite MoM having fewer total parameters. This strongly demonstrates that the performance gains stem from MoM's unique mixture and routing mechanism, rather than just from an increase in raw capacity or parameter count.
>
> | Model | Params | FDA | SWDE | SQUAD | NQ | TriviaQA | Drop | Avg. |
> | :--- | :--- | :--- | :--- | :--- | :--- | :--- | :--- | :--- |
> | GLA *expanded* | 425M | **15.08** | 20.15 | 28.28 | 13.30 | 41.65 | 18.74 | 22.87 |
> | GLA *MoM* | 395M | 9.90 | **21.65** | **29.36** | **14.16** | **45.20** | **20.89** | **23.53** |
> ||
> | Gated DeltaNet *expanded* | 550M | 18.26 | 24.27 | **30.03** | **17.74** | 48.34 | 19.26 | 26.32 |
> | Gated DeltaNet *MoM* | 444M | **22.98** | **29.90** | 29.69 | 16.60 | **48.82** | **20.99** | **28.16** |
>
> ---
>
> References:
>
> [2] Test-time regression: a unifying framework for designing sequence models with associative memory
>
> [3] Neural Network Capacity Using Delta Rule

---

> ### Author Response · Authors · 2025-11-17
> **[Part 3] Response to Reviewer vGVV**
>
> > `Question 3: Source of Gains in Hybrid Architectures.` Do the observed gains mainly result from increased depth, or is there measurable synergy between global attention and mixture-based memory modules?
>
> We first want to clarify that the observed gains do not result from increased depth. The hybrid architecture maintains the same 24-layer configuration as the baselines. In this setup, we simply replace one MoM layer with one full attention layer every 8 layers.
>
> The experimental results show that this hybrid architecture substantially outperforms both the pure MoM and pure Transformer models. We attribute this significant gain to the measurable synergy from mixing two distinct memory and learning mechanisms:
>
> 1. **Complementary Memory Types**: The MoM layers provide a global, compressed memory, while the Transformer layers provide precise, fine-grained, token-to-token memory. Combining them allows different parts of the model to specialize, focusing on either compressed global information or exact recall information, creating a powerful complementary effect.
> 2. **Complementary Learning Paradigms**: The two modules also operate on different learning principles. The MoM layers utilize a learning process based on the Delta Rule, while the Transformer layer can be seen as a form of non-parametric regression [2]. We hypothesize that this synergy between two different learning styles also contributes to the improved performance.
>
> We thank you again for your insightful questions! We hope these clarifications and new results have addressed your concerns and further demonstrated the merits of our work.
>
> ---
>
> References:
>
> [2] Test-time regression: a unifying framework for designing sequence models with associative memory

---

> ### Author Response · Authors · 2025-11-24
> **Thank you & Looking Forward to Further Discussion**
>
> Dear Reviewer vGVV,
>
> We would like to extend our heartfelt thanks for your time and effort in reviewing our work. Your perceptive comments, particularly regarding the router's implementation and the interpretability of memory specialization, have greatly inspired us to enhance our paper. As the discussion period progresses, we wish to ensure that our rebuttal has sufficiently addressed your concerns. For clarity, we have summarized our main responses below:
>
> - **Router Implementation & Gradients.** `Weakness 1 & Question 1` We clarified that gradients back-propagate through the selected Top-K paths via the weighted sum mechanism ($\tilde{\textbf M}_t$). We also detailed the auxiliary loss for load balancing, supported by the ablation study in Table 6.
> - **Quantitative Memory Specialization.** `Weakness 2` Inspired by your feedback, we conducted a rigorous quantitative UMAP analysis (Figure 5 in Revision). The results reveal clear clustering of tokens routed to specific memories in deeper layers, demonstrating memory specialization. We interpret this as a form of ensemble learning within the Test-Time Training (TTT) framework.
> - **Source of Improvement.** `Question 2` We presented comparative results in Table 4 showing MoM outperforms the single expanded memory baseline even with fewer parameters. This confirms the gains stem from the mixture-and-routing mechanism rather than mere capacity expansion.
> - **Hybrid Architecture Synergy.** `Question 3` We clarified that both the hybrid model and the baseline share the same number of layers. The observed performance boost is attributed to the synergy between complementary memory types and learning mechanisms.
>
> Please refer to our detailed rebuttal for points not covered in this short summary. We sincerely hope that these revisions and clarifications have addressed your concerns. Your constructive feedback has been invaluable to us, and we are thankful for the opportunity to improve our manuscript under your guidance. We humbly look forward to any further discussion with you.
>
> Warm regards,
>
> Authors

---

### Author Response · Authors · 2025-11-24
**General Response**

Dear Reviewers and ACs,

We commence by extending our sincere gratitude to all reviewers for their time and insightful comments. We are encouraged to see that the reviewers recognize the core strengths of our work, including the comprehensive evaluation and superior performance on recall-intensive tasks (Reviewers vGVV, EVGR, SvM8, 51kq) , the novel application of sparse activation to RNN states (Reviewer EVGR) , and the hardware-efficient implementation that maintains linear complexity (Reviewers vGVV, SvM8) . Your expertise has been instrumental in helping us refine the rigor and presentation of this paper.

We are particularly heartened by **Reviewer SvM8's decision to raise the score to 6**. We represent our deepest gratitude for this strong endorsement, which serves as a tremendous encouragement for our work.

We apologize for any inconvenience caused by the omission of certain details in the article and endeavor to respond to each comment. We sincerely hope that the responses can address the reviewers' concerns. For reference, we present a brief introduction of the response as follows.

- **Quantitative Analysis of Memory Specialization.** `Reviewers vGVV, EVGR, SvM8` Inspired by the reviewers' insightful suggestions, we replaced the original qualitative analysis with a more rigorous visualization analysis using **UMAP**, now presented in **Figure 5** of the Revision. The results reveal that tokens routed to different memories exhibit clear clustering phenomena, confirming the specialization of each memory. Furthermore, we interpret this mechanism from the novel perspective of **Test-Time Training (TTT)** as a form of Test-Time ensemble learning, where memories specialize in fitting specific sub-distributions of the data.

- **Source of Performance Gains.** `Reviewers vGVV, EVGR, SvM8` To address the fundamental question regarding the validity of MoM's mechanism, we conducted a control experiment in **Table 4** comparing MoM against a **single expanded memory** baseline. We selected two representative models with fundamentally different learning mechanisms: GLA (representing Hebbian-like learning rules) and Gated DeltaNet (representing the Delta Rule). We are pleased to report that MoM consistently outperforms the single expanded memory under **both mechanisms**. This validates that the performance gains originate from the innovative **mixture and separation mechanism** of MoM, rather than merely from shared memory, parameter count, or raw memory capacity.

- **Router Mechanics.** `Reviewers vGVV, SvM8, 51kq` We detailed the router's training process, which is driven by **gradients propagated** from the final task loss and an auxiliary load-balancing loss . From the perspective of linear RNN parallelism, we clarified the design choice of using a **state-independent** router to maintain high training efficiency with minimal computational overhead (<2%). Crucially, the distinct **clustering phenomena** observed in the UMAP visualization of routed tokens validates that this mechanism successfully learns a meaningful, specialized routing policy rather than performing random selection.

- **Hyperparameter Search.** `Reviewers SvM8, 51kq` In **Table 6**, **Figure 7**, and **Appendix H**, we thoroughly discussed the search for hyperparameters, including the number of memories, activation ratios, and the shared memory component. We clarified that the selection of the $N=4, k=2$ configuration represents a favorable trade-off derived from these extensive ablation studies, balancing model performance with computational efficiency.

- **Hybrid Architecture Synergy.** `Reviewer vGVV` We clarified that the hybrid model maintains the exact same depth (24 layers) as the baselines. Despite this equality in depth, it significantly outperforms both the pure MoM and Transformer baselines across various tasks. We attribute these substantial gains to the synergy between complementary **memory mechanisms** and **learning paradigms**.

- **Positioning and Writing Improvements.** `Reviewers EVGR, vGVV` We refined the narrative ensure a precise academic tone by reducing biological analogies and removing overemphasized claims regarding the distinction from MoE . Additionally, we relocated the fair parameter comparison table to the main body to prioritize rigorous evaluation.

Thank you again for your valuable feedback. We firmly believe that these revisions have significantly strengthened the paper, and that our approach offers fresh perspectives and meaningful technical contributions to the community.

Sincerely,

Authors

---

### Meta-Review · Area_Chair_AaSQ · 2026-01-07

**Summary:**

Reviewers praised the novel sparse routing applied to linear RNN states, extensive empirical gains on recall-intensive and long-context tasks, hardware-efficient linear-complexity implementation, and insightful memory analyses. The rebuttal effectively resolved most concerns via new controls (Table 4: MoM vs. expanded memory), UMAP specialization visualization (Figure 5), router gradient details, fair comparisons in main text, hyperparameter ablations, and moderated claims. Remaining requests for larger-scale experiments and recent baseline comparisons are valuable but non-blocking, as core contributions and empirical strength are well-established. Overall positive scores (one explicit increase from 4 to 6, others stable or upward) and resolved concerns justify acceptance.

**Reviewer Concerns:**

- Addressed: source of gains, quantitative specialization, router mechanics/gradients, parameter fairness, MoE distinction, shared vs. mixture role, hyperparameter search.
- Outstanding: larger-scale experiments; deeper comparisons to very recent baselines (e.g., RWKV-7, Titans).

**Reviewer Scores:**

- vGVV (initial 6): likely remains 6 or slight increase (concerns largely resolved).
- EVGR (initial 6): likely increases (positive on revisions).
- SvM8 (initial 4): raised to 6 (explicitly stated after rebuttal).
- 51kq (initial 6): likely remains 6 (minor concerns clarified).

---

### Decision · Program_Chairs · 2026-01-26

Accept (Poster)